# Comparing Uniform Price and Discriminatory Multi-Unit Auctions through Regret Minimization

**Marius Potfer**[1,2]     **Vianney Perchet**[1,3]

[1] Crest (Fairplay joint team), ENSAE
[2] EDF R&D
[3] Criteo AI Lab

## Abstract

Repeated multi-unit auctions, where a seller allocates multiple identical items over many rounds, are common mechanisms in electricity markets and treasury auctions. We compare the two predominant formats: uniform-price and discriminatory auctions, focusing on the perspective of a single bidder learning to bid against stochastic adversaries. We characterize the learning difficulty in each format, showing that the regret scales similarly for both auction formats under both full-information and bandit feedback, as $\tilde{\Theta}(\sqrt{T})$ and $\tilde{\Theta}(T^{2/3})$, respectively. However, analysis beyond worst-case regret reveals structural differences: uniform-price auctions may admit faster learning rates, with regret scaling as $\tilde{\Theta}(\sqrt{T})$ in settings where discriminatory auctions remain at $\tilde{\Theta}(T^{2/3})$. Finally, we provide a specific analysis for auctions in which the other participants are symmetric and have unit-demand, and show that in these instances, a similar regret rate separation appears.

## 1 Introduction

Uniform price auctions and discriminatory auctions are widely used market mechanisms for the allocation of resources. They have been widely implemented and studied for electricity markets (Elmaghraby & Oren, 1999; Hortaçsu & Puller, 2008) and treasury auctions (Nyborg et al., 2002). They are the most widespread simultaneous multi-unit auctions of identical goods (Krishna, 2009). Uniform price and discriminatory auctions are both standard mechanisms that allocate items to buyers willing to pay the most; their rules are identical except for the prices buyers pay. Since the two mechanisms are so similar, they can easily replace each other and have been frequently compared (De Keijzer et al., 2013). Their differences have been studied from several points of view: from their efficiency in Ausubel et al., 2014, the revenue they generate for the auctioneer, to the expected social welfare in different types of equilibria (Syrgkanis & Tardos, 2013).

Participating in these auctions can be challenging from a bidder's standpoint since the bidding process is not truthful (i.e., the best strategy depends on other buyers' behavior). However, when auctions occur repeatedly, buyers can improve their strategies over time using online learning techniques; this is referred to as *learning to bid* (Feng et al., 2018). The performance of these online learning strategies is often measured by regret, which compares the cumulative rewards with that of the best fixed strategy in hindsight (Lattimore & Szepesvári, 2020). We propose to leverage this online learning framework to quantify the difficulty of learning to bid optimally, therefore providing a new approach to compare uniform and discriminatory price auctions.

Learning to bid in repeated multiunit auctions has recently been studied, both in the uniform pricing case Brânzei et al., 2023 and Potfer et al., 2024, and in the discriminatory pricing case by Galgana and Golrezaei, 2025. These works focus on adversarial bids and provide regret rates with similar

39th Conference on Neural Information Processing Systems (NeurIPS 2025).

dependency on the time horizon $T$ of $\tilde{\mathcal{O}}(\sqrt{T})$ under full-information feedback and $\tilde{\mathcal{O}}(T^{2/3})$ under bandit feedback. These rates have been proven to be tight, except for the usual uniform price auction (referred to as the First Rejected Bid in Potfer et al., 2024).

We study the learning-to-bid problem in the case where opposing bids are stochastic. The stochasticity assumption ensures that our analysis characterizes the inherent difficulty of the auction format, rather than the difficulty arising from facing strategic or adversarial agents. We characterize when both problems can be learned with similar regret rates and when the rates differ. In the process, we provide the first worst-case tight lower bounds for uniform price auctions with bandit feedback, as well as the first instance-dependent regret bounds. Finally, we show that, against symmetric unit-demand adversaries, a clear separation of achievable regret appears, and we provide an efficient algorithm for the uniform auction that leverages the specific structure of this setting.

## 1.1 Related Work

Simultaneous auctions of multiple identical items, including uniform, discriminatory, and Vickrey-Clarke-Groves pricing, are standard in auction theory, as described in Krishna, 2009. Uniform and discriminatory auctions have been studied and compared from an empirical point of view (Nyborg & Sundaresan, 1996; Nyborg et al., 2002), and by theoretical approaches, focusing on the revenue they generate (Ausubel et al., 2014) as well as the social welfare they generate (De Keijzer et al., 2013; Syrgkanis & Tardos, 2013). Special cases involving symmetry and unit-demand participant are also of particular interest (Anderson & Holmberg, 2023).

The repeated setting of auctions has recently attracted some focus, as covered by Nedelec et al., 2022. This setting allows for leveraging online and statistical learning tools (Lattimore & Szepesvári, 2020) to explore dynamic strategies. These repeated settings were first studied with a focus on the auctioneer, enabling the learning of reserve prices as in the work of Mohri and Medina, 2014. Learning to bid, from the bidder's perspective, was introduced later and studied in several settings, including sealed-bid first-price and second-price auctions (Achddou et al., 2021b; Balseiro et al., 2019; Weed et al., 2016).

Learning to bid in multi-unit auctions has only recently started to be studied, except for a partial result for uniform auctions by Feng et al., 2018 used as an example. The discriminatory pricing auction was studied by Galgana and Golrezaei, 2025, who proved regret rates of $\tilde{\mathcal{O}}(K\sqrt{T})$ and $\tilde{\mathcal{O}}(KT^{2/3})$ under full-information and bandit feedback, respectively. Regret bounds of $\tilde{\mathcal{O}}(K^{3/2}\sqrt{T})$ were also obtained for the uniform-price format in the full-information case by Brânzei et al., 2023, as well as sub-optimal rates in the bandit case. Potfer et al., 2024 provided algorithms with a $\tilde{\mathcal{O}}(K^{4/3}T^{2/3})$ regret upper bound for uniform auction in bandit feedback and showed that it is tight for a less common variant of uniform pricing (Last Accepted Bid pricing, or LAB). They provide no matching lower bound for the usual uniform price auction (First Rejected Bid, or FRB). Golrezaei and Sahoo, 2024 has also studied uniform pricing in a repeated setting, but focuses on bidders with return-over-investment constraints; they obtain similar regret bounds of $\tilde{\mathcal{O}}(K^{5/3}T^{2/3})$ in their setting.

## 1.2 Contribution

We propose a regret-based comparison of repeated uniform and discriminatory multi-unit auctions with stochastic opposing bids. For this comparison, we provide the first analysis of learning to bid in repeated multi-unit auctions when facing stochastic bids for both auction types.

We show that both auction formats admit tight worst-case regret rates of $\tilde{\mathcal{O}}(K\sqrt{T})$ under the full-information setting (improving the best known rates for uniform price auction by a factor of $\sqrt{K}$). We provide the first lower bound for a uniform price auction designed specifically for bandit feedback, showing that the regret must grow as $\Omega(T^{2/3})$, which, in light of previously known upper bounds (Potfer et al., 2024), fully characterizes how the regrets must scale with respect to $T$. Our algorithm analysis also yields a more precise regret characterization for the bandit setting of uniform price auctions: while the upper bound scales as $\tilde{\mathcal{O}}(K^{5/3}T^{2/3})$ for general demand, better regret rates can be achieved when the bidder has low demand: 0 regret for unit-demand and $\tilde{\mathcal{O}}(\sqrt{T})$ for two-unit-demand.

We further provide an instance-dependent quantity characterizing other instances (with general demand) for which regret bounds scaling as $\mathcal{O}(\sqrt{T})$ can be obtained for the uniform auction under

bandit feedback. The discriminatory auction's lower bound of $\Omega\left(T^{2/3}\right)$ remains valid in these instances.

Finally, in the bandit feedback when the adversaries are unit-demand and symmetric we show a similar regret separation providing an algorithm which guarantees regret of $\tilde{\mathcal{O}}(\sqrt{T})$ in uniform auctions while lower bound of $\Omega\left(T^{2/3}\right)$ for the discriminatory price auctions remains valid. In the analysis, we derive a concentration inequality for partially observed ordered statistics based on the DKW inequality, which may be of independent interest.

The following table summarizes the achievable regret rates for the different auction formats under bandit feedback considered in this work, we use the notation $\tilde{\Theta}(\cdot)$ when we have matching lower bounds $\Omega(\cdot)$ and upper bounds $\mathcal{O}(\cdot)$, up to log factors. Results that are entirely novel are highlighted in green, while blue denotes marginal improvements or generalizations over existing work.

|  | **Worst Case** | | | **Instance Dependent** | |
|---|---|---|---|---|---|
|  | Unit Demand | Two-unit Demand | Any Demand | $\Delta$- sep 4 | I.I.D. |
| Discriminatory | $\tilde{\Theta}(T^{2/3})$ | | | $\tilde{\Theta}(T^{2/3})$ | |
| Uniform | 0 | $\tilde{\Theta}(\sqrt{T})$ | $\tilde{\Theta}(T^{2/3})$ | $\tilde{\Theta}(\sqrt{T})$ | |

Table 1: Achievable regret under bandit feedback.

## 1.3 Technical novelty and Challenges

While we study a similar setting, our approach departs significantly from prior work (Potfer et al., Branzei et al., Galgana et al.) in both algorithmic design and theoretical analysis. Unlike existing methods that discretize the bid space and rely on randomized bandit algorithms, we propose a deterministic algorithm tailored to the stochastic opposing bids setting. The main technical novelty lies in the use of statistical CDF estimation and concentration bands, which allow us to operate directly in a continuous action space without discretization. This analysis yields a cleaner and more interpretable analysis. From a technical standpoint, the main challenges is to carefully choose and apply the right concentration bounds and results to ensure the tightness of our bounds.

We establish two complementary lower bounds that characterize the fundamental limits of learning in these stochastic auctions. The first provides a general lower bound that extends prior discrete-space results to continuous bid domains, ensuring that the regret scaling remains valid regardless of the action granularity. More importantly, deriving the lower bound for uniform price auctions under bandit feedback required particular care. As shown in Lemma 2, the feedback is richer than standard bandit feedback (since it includes local observation of opposing bids), making it necessary to finely manage the feedback available to obtain our tight lower bound.

## 2 Models

The following encompasses both uniform and discriminatory price multi-unit auctions, and matches those used in Brânzei et al., 2023 and Galgana and Golrezaei, 2025. Indeed, because both auction formats are very similar, we only need to specify different payment rules (equations (2) and (3)) to describe the mechanisms.

**Repeated auction**   Let $(K, T) \in \mathbb{N}^2$ be, respectively, the number of items sold at each auction and the number of repeated auctions. We describe the repeated $K$-item uniform and discriminatory price auctions from the point of view of one buyer, that we call the *bidder*, facing other stochastic buyers, whose bids are aggregated and who, as a group, are called the *adversary*. The bidder has a fixed set of valuations that quantifies how much utility it derives from being allocated any number of items. We denote by $(v_1, ..., v_K) \in [0, 1]^K$ the bidder's marginal valuations, they are assumed to be non-increasing. These are marginal in the sense that when the bidder obtains $n$ items, it derives $\sum_{l \leq n} v_l$ from it.
At each time-step $t \in [T]$, an instance of the auction proceeds as follows:

1. The bidder transmits its bids $\mathbf{b}^t := \{b_1^t, b_2^t, ..., b_K^t\}$ to the auctioneer. For convenience, we consider $\mathbf{b}$ ordered in non-increasing order and the bids in $[0, 1]^K$. We call $B$ the corresponding subset of $[0, 1]^K$.

2. The bids of the adversary $\boldsymbol{\beta}^t := \{\beta_1^t, \beta_2^t, ..., \beta_K^t\} \in B$ are drawn according to a distribution over $B$, denoted $\mathcal{D}$, and then transmitted to the auctioneer. Note that since opposing bids belong to $B$, they are also assumed to be already sorted in non-increasing order.
3. The auctioneer orders all the bids in non-increasing order and determines the *allocation* (the number of items won) of the bidder that we denote $x\left(\mathbf{b}^t, \boldsymbol{\beta}^t\right) \in [K]$ as well as the prices $p\left(\mathbf{b}^t, \boldsymbol{\beta}^t\right) \in [0,1]^K$.
4. The bidder receives their allocated items and pays the price for each item. This gives rise to the following utility: $u\left(\mathbf{b}^t, \boldsymbol{\beta}^t\right) := \sum_{l=1}^{x(\mathbf{b}^t, \boldsymbol{\beta}^t)} [v_l - p(\mathbf{b}^t, \boldsymbol{\beta}^t)(l)]$
5. The bidder may receive additional information, referred to as the feedback.

**Allocation**   Both auctions we study are standard; therefore, the items are allocated to the buyers who submitted the $K$ highest bids. Thus, the allocation is formulated as follows[1]:

$$x(\mathbf{b}, \boldsymbol{\beta}) := \max\{k \in [K] \mid \forall j \leq k, b_j \geq \beta_{K+1-j}\} \tag{1}$$

**Pricing**   As we noted previously, the pricing rules of the uniform and discriminatory auctions are different. We describe below how the price is set in both auctions:

- The uniform price auction sets the price of all allocated units identically: the value of the first rejected bid. Formally, it is the $(K+1)^{\text{th}}$ largest element of $(\mathbf{b}, \boldsymbol{\beta})$, written:

$$p(\mathbf{b}, \boldsymbol{\beta}) := \big(\max(b_{x(\mathbf{b}, \boldsymbol{\beta})+1}, \beta_{K-x(\mathbf{b}, \boldsymbol{\beta})+1})\big)_{k \in [K]]} \tag{2}$$

- The discriminatory price auction sets the price of each allocated unit as the value of the corresponding bid (the bid that won its emitter this item). Formally, the discriminatory price is written as follows:

$$p(\mathbf{b}, \boldsymbol{\beta}) := (b_k)_{k \in [K]} \tag{3}$$

**Regret**   The performance of a learning algorithm is quantified by the regret. This metric quantifies the expected cumulative difference between the maximum utility the bidder could have obtained and the utility they actually received. We define it as follows:

$$R_T = T \sup_{\mathbf{b} \in B} \left( \underset{\boldsymbol{\beta} \sim \mathcal{D}}{\mathbb{E}} [u(\mathbf{b}, \boldsymbol{\beta})] \right) - \sum_{t=1}^{T} \underset{\boldsymbol{\beta}^t \sim \mathcal{D}}{\mathbb{E}} [u(\mathbf{b}^t, \boldsymbol{\beta}^t)] \tag{4}$$

This definition aligns with what is usually defined as pseudo-regret. Note that, with the previous definition, minimizing the regret is equivalent to maximizing the cumulative expected utility. Using this metric allows us to take into account both how well and quickly a bidder can approximate the best bid, as well as how *costly* it is to acquire information about $\mathcal{D}$.

**Feedback**   The bidder can sequentially improve its bids since it receives some information after each auction, which intuitively provides it with some knowledge about the distribution $\mathcal{D}$. We describe below the *feedback*, defined as the information received after one instance of the auction. We focus on two feedback types: the full-information and bandit feedback setting, which are the most common in online learning literature (Lattimore & Szepesvári, 2020) and in online learning in auctions (Achddou et al., 2021a; Feng et al., 2018). At time $t \in [T]$, at the end of the auction:

- in the *full-information* feedback setting, the bidder observes $\boldsymbol{\beta}^t$;
- in the *bandit* feedback setting, the bidder observes only $x(\mathbf{b}^t, \boldsymbol{\beta}^t)$ and $p(\mathbf{b}^t, \boldsymbol{\beta}^t)$.

Note that the full-information feedback is strictly more informative than the bandit feedback, because $x(\mathbf{b}^t, \boldsymbol{\beta}^t)$ and $p(\mathbf{b}^t, \boldsymbol{\beta}^t)$ can be computed by the bidder who knows both its bids $\mathbf{b}^t$ and the auction's rules.

## 3   Learning with full-information feedback

We first focus on the full information feedback setting where the learner observes the full vector $\boldsymbol{\beta}^t$ at each time step. The bidder aims to minimize regret by choosing bids as close as possible to the optimal bid vector, namely $\mathbf{b}^\star := \arg\max_{\mathbf{b} \in B} \mathbb{E}_{\boldsymbol{\beta} \sim \mathcal{D}}[u(\mathbf{b}, \boldsymbol{\beta})]$. Because the auction mechanisms

---

[1]This allocation breaks ties in favor of the bidder. The results can be extended to any tie-breaking rule by ensuring that ties occur with probability zero.

are not truthful, $\mathbf{b}^\star$ depends on $\mathcal{D}$. To describe this dependence, let us define, for all $k \in [K]$, and $x \in [0, 1]$, $F_k(x) := \mathbb{P}_{\boldsymbol{\beta} \sim \mathcal{D}}(\beta_k \leq x)$ the $k^{th}$ marginal cumulative distribution function.

The following lemma provides the main intuition for our learning algorithms, which mainly focus on estimating the marginal CDFs rather than the optimal bid vector $\mathbf{b}^\star$.

**Lemma 1.** *In both auction formats, the expected utility can be expressed as a function of the bidder's bid vector $\mathbf{b}$ and the marginal cumulative distribution functions $(F_k)_{k \in [K]}$.*

The proof follows from writing the utility and taking the expectation, it is provided in Appendix A. We denote $U_d$ the functions such that, in the discriminatory auction, for all $\mathbf{b} \in B, U_d((F_k)_{k \in [K]}, \mathbf{b}) = \mathbb{E}_{\boldsymbol{\beta} \sim \mathcal{D}}[u(\mathbf{b}, \boldsymbol{\beta})]$, and $U_u$ the equivalent in the uniform auction. Explicit formulas are provided in the appendix (see (14) and (15)) to save space.

### 3.1 Algorithm

At time $t \in [T]$, the bidder has observed $\boldsymbol{\beta}^1, \ldots, \boldsymbol{\beta}^{t-1}$, which are i.i.d. samples from $\mathcal{D}$. These are sufficient to compute empirical marginal CDFs, which are known to have strong convergence properties Massart, 1990. They are defined, for $k \in [K]$, and $x \in [0, 1]$ as $\hat{F}_k^t(x) := \frac{1}{t-1} \sum_{j=1}^{t-1} \mathbb{1}\{\beta_k^j \leq x\}$. Depending on the auction, we define the expected utility estimate as either $\hat{u}^t(\mathbf{b}) := U_u((\hat{F}_k^t), \mathbf{b})$ or $\hat{u}^t(\mathbf{b}) := U_d((\hat{F}_k^t), \mathbf{b})$. Equipped with these, we can now provide an algorithm that guarantees tight regret bounds for both auction formats.

---

**Algorithm 1:** Full-information feedback

---

1 **Input:** time horizon $T$
2 **Output:** bids for each time step $(\mathbf{b}^1, \mathbf{b}^2, \ldots, \mathbf{b}^{T-1}, \mathbf{b}^T) \in (B)^T$.
3 **for** $t = 1, 2, \ldots, T$ **do**
4 $\quad$ Play $\mathbf{b}^t := \mathrm{argmax}_{\mathbf{b} \in B} \, \hat{u}^t(\mathbf{b})$.
5 $\quad$ Receive the utility $u(\mathbf{b}^t, \boldsymbol{\beta}^t)$ and observe $\boldsymbol{\beta}^t$.

---

**Theorem 1.** *Under full-information feedback, Algorithm 1 achieves a regret $R_T = \tilde{\mathcal{O}}\left(K\sqrt{T}\right)$ for both the discriminatory and the uniform price auction.*

*Proof Sketch.* The proof for the discriminatory auction uses the Dvoretzky–Kiefer–Wolfowitz inequality from Massart, 1990 to build high-probability confidence bands for $\hat{F}_k$, which yields confidence bands for the utility estimate. Combining that with the optimality of $\mathbf{b}^t$ with respect to $\hat{u}^t$ allows us to show that the per-round regret is upper bounded by $2K\sqrt{\ln(2/\alpha)/2t}$, therefore providing the correct regret rates when summing.

The proof for the uniform auction's is very similar. The main difference appears when ensuring error bands scales as $K$ and not $K^2$, this requires our concentration bnads to leverage the strong negative correlation between the different terms estimated in excepted utility. This is achieved by using the Natarajan dimension, a multi-class classifier generalization of VC dimension Shalev-Shwartz and Ben-David, 2014. $\qquad\square$

The complete proof is in Appendix B. These regret bounds are tight, as it is known that any algorithm must incur a regret growing as $\Omega(K\sqrt{T})$ in both auction format (Brânzei et al., 2023; Galgana & Golrezaei, 2025).

**Remark 1.** *Note that, because $\hat{u}^t$ is always a piece-wise linear function and is separable into functions that only depend on two components of $\mathbf{b}$, a maximizer can be found by using dynamic programming in $\mathcal{O}\left(K^3 t^2\right)$.*

## 4 Learning with bandit feedback

We now focus on learning in the bandit feedback setting. Let us recall that in this setting, at time $t$, the bidder only observes his allocation $x(\mathbf{b}^t, \boldsymbol{\beta}^t)$ and the price paid per unit $p(\mathbf{b}^t, \boldsymbol{\beta}^t)$. Since the allocation function is the same across both auction's type (1), the observed allocation provides the same information :

$$\left(\mathbb{1}\left\{b_i^t \geq \beta_{K-i+1}^t\right\}\right)_{i \in [K]} \tag{5}$$

However, the information conveyed by the price differs:

- The discriminatory price depends only on the bidder's own bid and therfore provides no additional information.
- The uniform price can be set by an opposing bid $\beta_{K-i+1}$, potentially revealing information about the distribution $\mathcal{D}$.

The following lemma formalizes the feedback the bidder receives in the uniform price auction.

**Lemma 2.** *At time $t \in [T]$, let $\mathbf{b}^t, \boldsymbol{\beta}^t \in B$ be the bids of the learner and the adversary, in the uniform auction with bandit feedback the bidder observes $\left(\mathbb{1}\left\{\beta^t_{K-i+1} \in (b^t_{i+1}, b^t_i]\right\} \beta^t_{K-i+1}\right)_{i \in [K]}$.*

This feedback reveals components of $\boldsymbol{\beta}^t$ if they fall into the right intervals, providing richer feedback than in the discriminatory case. We now examine achievable regret rates in both auction types and the effects of this feedback discrepancy.

## 4.1 Discriminatory price auction

The discriminatory price auction with fixed valuation has been studied by Galgana and Golrezaei, 2025 who provided algorithms guaranteeing regret upper bounds of $\mathcal{O}\left(KT^{2/3}\right)$. They also provided a regret lower bound of $\Omega\left(K^{2/3}T^{2/3}\right)$ for discretized bid strategies. We provide a strengthening of this result with a lower bound that holds for *any* algorithm, including those operating over continuous bid spaces.

**Lemma 3.** *Any algorithms for bidding in repeated first price auctions with known valuation must incur a regret of $\Omega\left(T^{2/3}\right)$*

Given this result and the almost matching upper bound, we believe the achievable regret rates are well characterized. In the following we characterize what can be achieved in the uniform price auction, and when the richer feedback allows for better rates than in discriminatory auctions.

## 4.2 Uniform price auction

The additional information provided by the richer bandit feedback in the uniform price auction can be used to design learning algorithms. We illustrate how the feedback allows to estimate the marginal CDFs $F_k$ with Algorithm 2. During its estimation phase, it leverages the feedback detailed in Lemma 2 to ensure observation of each $\beta_k$ in a round robbin fashion. Once the estimated CDFs are precise enought, an estimated best bid $\mathbf{b}^{T_{expl}}$ is computed and then played repeatedly during the exploitation phase.

We define the empirical CDFs for the uniform price auction with bandit feedback as follows for $k \in [K]$:

$$\forall x \in [0,1], \tilde{F}_k(x) := \frac{\sum_{j=1}^t \mathbb{1}\left\{x \in (b^t_{K+2-k}, b^t_{K+1-k}]\right\} \mathbb{1}\left\{\beta^t_k \le x\right\}}{\sum_{j=1}^t \mathbb{1}\left\{x \in (b^t_{K+2-k}, b^t_{K+1-k}]\right\}} \tag{6}$$

We also define the empirical expected utility $\tilde{u}$, for $\mathbf{b} \in B$ as $U_u((\tilde{F})_{k \in [K]}, \mathbf{b})$. With these we can provide our algorithms and the associated regret guarantees.

---

**Algorithm 2:** Estimate then commit for uniform price auction

---

1 **Input:** time horizon $T$, $T_{expl}$ duration of the exploration.
2 **Output:** bids for each time step $\left(\mathbf{b}^1, \mathbf{b}^2, \dots, \mathbf{b}^{T-1}, \mathbf{b}^T\right) \in B^T$.
3 **Estimate: for** $t = 1, 2, \dots, T_{expl}$ **do**
4      $k \leftarrow T - K\lfloor T/K \rfloor + 1$.
5      Play $\mathbf{b}^t$ such that $\forall i \le k, b^t_i = 1$ and $\forall i > k, b^t_i = 0$.
6      Receive the utility $u^t = u(\mathbf{b}^t, \boldsymbol{\beta}^t)$ and the feedback.
7 **Commit: for** $t = T_{expl} + 1, \dots, T$ **do**
8      Play $\mathbf{b}^t = \text{argmax}_{\mathbf{b} \in B}\ \hat{u}^{T_{expl}}(\mathbf{b})$

---

**Theorem 2.** *The estimate then commit algorithms, with the exploration time $T_{expl} = K^{2/3}T^{2/3}$ achieves a regret bound of $\tilde{\mathcal{O}}\left(K^{5/3}T^{2/3}\right)$*

*Proof Sketch.* The proof separates the analysis of the regret incurred during the exploration phase and the commitment. During the exploration phase it is no bigger than $K^{5/3}T^{2/3}$. The regret of the exploitation phase can be upper bounded by $K^{5/3}T^{2/3}$; obtaining this bound requires the use of variants of the DKW concentration inequality, which exhibit local dependencies as in Bartl and Mendelson, 2023; Blanchard and Voracek, 2024. A simpler analysis, using DKW inequalities (Massart, 1990) yields a looser bound of $K^2 T^{2/3}$ when setting $T_{expl} = KT^{2/3}$. $\qquad\square$

These regret guarantees match the rates in $T$ of the ones for the discriminatory price auction. Since the regret rates in the discriminatory auction are tight in $T$, it is natural to ask whether that is also true for the uniform price auction. Theorem 3 provides a matching lower bound, proving that a regret that grows as $T^{2/3}$ is optimal.

**Theorem 3.** *In the uniform price auction, any algorithms must incur a regret of $\Omega\left(T^{2/3}\right)$ for learning to bid with known valuations, when valuations are such that $v_3 > 0$.*

*Proof idea.* The proof of this theorems relies on standard approach (Cesa-Bianchi et al., 2024; Kleinberg & Leighton, 2003) to create regret lower bounds in bandit setting with continuous action space by embedding into our auction model a hard instance which is akin to a multiarmed bandit problem with $\mathcal{O}\left(T^{1/3}\right)$ arms. The main challenge is that the bandit feedback in the uniform price auction is relatively rich, as pointed out in Lemma 2. We overcome this difficulty with the two following techniques, for a specific $i \in [K]$:

- the optimal bids are such that $b_i^\star = b_{i+1}^\star$ while the bidder needs to observe $\beta_{K-i+1}$,

- when submitting a bid with $b_i \neq b_{i+1}$, the bidder suffers an additional regret term which scales linearly with $b_i - b_{i+1}$.

$\qquad\square$

We have shown that the achievable regret in the two auction formats behaves similarly regardless of the feedback types. The remainder of the paper is dedicated to showing that, beyond worst cases, the uniform auction can be easier to learn (regret scaling with $\sqrt{T}$) across different families of instances for which the discriminatory auctions remain equally hard (regret scaling as $T^{2/3}$). We provide general algorithms that exhibit better regret scaling when possible without information about the instance and then focus on an I.I.D setting, whose structure is especially interesting.

## 5 Bandit feedback, beyond worst-case

### 5.1 Instance dependant regret

To exhibit that the regret scales as $\sqrt{T}$ on family of instances that are well-suited for the uniform price auction, we need an algorithm which adapts to such instances. This adaptability is key in guaranteeing worst-case regret, as in Theorem 2, while also leveraging easy instances when presented with one. Algorithm 2 lacks such flexibility, because of both its fixed-length estimation and commitment phases, as well as its fixed marginal CDF estimation intervals (namely $[0, 1]$).

We propose two natural improvements: $(i)$ using sequentially shrinking estimation intervals and $(ii)$ using distinct intervals for each marginal CDF. Intuitively, the shrinking rule should allow us to estimate CDFs only However, the choice of the shrinking rule must be made with care. We propose a successive elimination approach (Audibert & Bubeck, 2010) which iteratively reduces a set of candidate bids $B^t \subset B$ based on the current utility estimates and their uncertainty. As the set of relevant bids $B^t$ gets smaller, the intervals where estimates of the marginal CDFs are useful to estimate the utility do as well; this yields our shrinking rules for the intervals, that we denote $(\mathcal{I}_k^t)_{k \in [K]}$.

---

**Algorithm 3:** Bandit Feedback

---

**Input:** Time horizon $T$
**Initialize:** For each $i \in [K]$, initialize $\mathcal{I}_i^0 \leftarrow [0,1]$
**Output:** Bids $(\mathbf{b}^1, \ldots, \mathbf{b}^T) \in B^T$, with $\Delta^0 \leftarrow -1$

1 **for** $t = 1, \ldots, T$ **do**
2     **if** $t \leq T_{expl}$ **and** $\Delta^{t-1} \leq 0$ **then**
3        $k \leftarrow T - K\lfloor T/K \rfloor + 1$ ;
4        $b_k^t \leftarrow \max\{b_k \in \mathcal{I}_k^{t-1}\}, \quad b_{k+1}^t \leftarrow \min\{b_{k+1} \in \mathcal{I}_{k+1}^{t-1}\}$ ;
5        $\mathbf{b}^t \leftarrow \arg\max_{\substack{\mathbf{b} \in B \\ b_k = b_k^t,\, b_{k+1} = b_{k+1}^t}} \tilde{u}^t(\mathbf{b})$ ;
6     **else**
7        $\tilde{k} \leftarrow T - 2(K-1)\lfloor T/2(K-1) \rfloor + 2, \quad k \leftarrow \lfloor \tilde{k}/2 \rfloor$ ;
8        $b_k^t \leftarrow \begin{cases} \max\{b_k \in \mathcal{I}_k^{t-1}\} & \text{if } k' \text{ is even} \\ \min\{b_k \in \mathcal{I}_k^{t-1}\} & \text{if } k' \text{ is odd} \end{cases}$ ;
9        $\mathbf{b}^t \leftarrow \arg\max_{\substack{\mathbf{b} \in B \\ b_k = b_k^t}} \tilde{u}^t(\mathbf{b})$ ;
10     Play $\mathbf{b}^t$ and receive feedback ;
11     Update $(\mathcal{I}_k^t)_{k \in [K]}, \Delta^t$ using Algorithm 4 ;

---

---

**Algorithm 4:** Interval Refinement

---

**Input:** Time $t$, horizon $T$, intervals $(\mathcal{I}_k^{t-1})_{k \in [K]}$, utility estimate $\tilde{u}^t$
**Output:** Updated intervals $(\mathcal{I}_k^t)_{k \in [K]}$, gap $\Delta^t$

1 $u_{\max}^t \leftarrow \max_{\mathbf{b} \in \prod_{k \in [K]} \mathcal{I}_k^{t-1}} \tilde{u}^t(\mathbf{b})$ ;            `// Max observed utility`

2 $\prod_{k \in [K]} \mathcal{I}_k^t \leftarrow \text{conv}\left( U^{-1}\left( \left[ u_{\max}^t - \sqrt{\frac{\ln(2T^2)}{2\lfloor t/K \rfloor}}, u_{\max}^t \right] \right) \right)$ `// `$U^{-1}$` denote the preimage of `$\tilde{u}^t$

3 $\Delta^t \leftarrow \min_{k \in \{2,\ldots,K\}} \left( \min \mathcal{I}_k^t - \max \mathcal{I}_{k+1}^t \right)$ ;

---

Let us denote $B^\star \subset B$ the set of maximizers of the expected utility and the optimal intervals $\mathcal{I}_k^\star \subset \mathbf{R}$ the smallest closed intervals such that $B^\star \subset \prod_{k=1}^K \mathcal{I}_k^\star$. We introduce the interval gap $\Delta := \min_{k \in \{2,\ldots,K\}} \left( \min \mathcal{I}_k^\star - \max \mathcal{I}_{k+1}^\star \right)$, the following theorem provides instance dependant bounds, valid when $\Delta > 0$ ( $\mathcal{O}(\cdot)$ hides terms constant in $T$).

**Theorem 4.** *Algorithm 3 guarantee a regret of* $\tilde{\mathcal{O}}(K^{5/3}T^{2/3})$ *in general and, when* $\Delta > 0$, *an instance-dependent regret bounds of* $\tilde{\mathcal{O}}(K\sqrt{T})$.

*Proof.* Worst-case regret bounds are essentially proved as in Theorem 2. We only need to check that our shrinking rule is well-behaved. Noticing that with high probability, the intervals $\mathcal{I}_i^t$ always contain the best-response set $B^\star$ suffices.

Instance-dependent bounds proofs rely on the fact that as soon as intervals become disjoint (ie $\Delta^t > 0$), the observation of Lemma 2 covers the whole intervals, which is why we recover full-information type bounds. $\qquad \square$

### 5.2 Regret separation

We illustrate the implications of Theorem 4 by describing instance families for which the achievable regret rates in uniform and discriminatory price auctions diverge significantly. This highlights how the choice of auction impacts learning complexity.

A straightforward example is when the bidder has *unit demand*, meaning it only values a single unit ($v_1 > 0$ and $v_i = 0$ for $i > 1$). In this case, the uniform auction is truthful (Lemma 7), so the bidder can bid truthfully and incur zero regret. In contrast, the discriminatory auction still requires strategic bidding, and the lower bound from Lemma 3 applies, resulting in a regret of $\Omega\left(T^{2/3}\right)$.

**Two unit demand** A similar but weaker regret rate discrepancy also appears in the two-unit demand setting (i.e. $v_1 > 0, v_2 \geq 0$ and $v_i = 0$ for $i > 2$). In that setting, both auctions require strategic

bidding; in discriminatory auction, Lemma 3 still applies resulting in regret $\Omega(T^{2/3})$ while Theorem 4 yields a regret of $\mathcal{O}(\sqrt{T})$ on the regret in the uniform auction.

**$\Delta$ separated distributions**    We finally consider a more general condition on the distribution of adversary bids. Let $\mathcal{D}$ be a distribution over $B$ such that when $\boldsymbol{\beta} \sim \mathcal{D}$ each coordinate $\beta_k$ almost surely lies in disjoint intervals $\mathcal{I}_k$, and adjacent intervals are at least separated by a gap $\Delta$. We refer to these as $\Delta$ separated. To rule out degenerate cases, we focus on $\Delta < \frac{1}{2K}$:

**Lemma 4.** *Learning in a discriminatory or uniform auction when the adversary's distribution is $\Delta$ separated, with $\Delta < \frac{1}{2K}$ can be achieved with respectively regret of $\Omega(T^{2/3})$ and $\mathcal{O}(\sqrt{T})$.*

*Proof sketch.* The lower bound from Lemma 3 for the discriminatory auction can be easily extended by leveraging the fact that $\Delta > \frac{1-v_1}{K}$. Proving the upper bound is straightforward by noticing the following : (i) there is always an optimal bids against $\Delta$-separated distribution whose component are only 0s and 1s, (ii) the $\bar{F}$ of $\Delta$-separated distribution are $\Delta$-separated. $\qquad\square$

### 5.3   I.I.D. adversaries

This section characterizes achievable regret when the bidder faces $N \in \mathbb{N}$ symmetric, unit-demand participants. Each participant submits a single bid, so the opposing bid vector $\boldsymbol{\beta}$ consists of the $K$ highest bids among them. By symmetry, all bids are i.i.d. from a distribution $\mathcal{P}$, and we assume independence in their bidding. We denote by $\mathcal{P}_K^N$ the induced distribution on $\boldsymbol{\beta}$.

**Opposing bids**    The first $K$ order statistics of $N$ i.i.d. samples from $\mathcal{P}$ constitute the opposing bids. This structure provides additional information on the relationships between marginal CDFs (Casella & Berger, 2024). Letting $F : [0, 1] \to [0, 1]$ denote the CDF of $\mathcal{P}$, we have for all $k \in [K]$:

$$F_k(x) := \mathbb{P}_{\boldsymbol{\beta} \sim \mathcal{P}_K^N} (\beta_k \leq x) = F^{N-K}(x) \sum_{j=1}^{k} \binom{N}{N-j} F^{K-j+1}(x) (1 - F(x))^{j-1} \qquad (7)$$

Note that this is a polynomial in $F$; in the following, we denote it $P_k$.

**Concentration Inequalities for order statistics**    Exploiting (7), we build cdf bands for all order statistics based on observations of a single order statistic. For simplicity, it is stated in the full information feedback. For $k, k' \in [K]^2$ define the empirical estimate of the CDF of the $k'^{\text{th}}$ order statistics based on the $k^{\text{th}}$: $\hat{F}_{k \to k'}^t := P_{k'}(P_k^{-1}(\hat{F}_k^t))$

**Lemma 5.** *Let $t, k \in \mathbb{N}$, and let $X_j^i$ for $i \in [t], j \in [k]$ be i.i.d. samples from a distribution $\mathcal{P}$ with cdf $F$. With probability $1 - \alpha$ :*

$$\hat{F}_{k \to k'}^t(x) - \alpha_{k,k'}\epsilon \leq F_{k'}(x) \leq \hat{F}_{k \to k'}^t(x) + \alpha_{k,k'}\epsilon \qquad (8)$$

*Where $\epsilon = \sqrt{\frac{\ln\left(\frac{2}{\alpha}\right)}{2t}}$ and $\alpha_{k,k'}$ is a constant which only depends on $k, k'$ and $N$.*

*Proof.* The lemma is proved using the DKW inequality (Massart, 1990), the order statistics cdf formula (7) and by noticing that all $P_k$ are strictly increasing on $(0, 1)$. $\qquad\square$

**Algorithm**    These concentration inequalities effectively enable recovery of full-information feedback in the uniform price auction by selecting the best estimate. For $k \in [K]$ and $x \in [0, 1]$, let $t_k(x)$ be the number of times $\mathbb{1}\{\beta_k \leq x\}$ was observed (Lemma 2). Define $k^\star(x) = \arg\max_{k \in [K]} t_k(x)$ as the index with the most observations, and estimate marginal CDFs as $\bar{F}_k^t(x) := \tilde{F}_{k^\star(x) \to k}^t(x)$. The resulting utility estimate is $\bar{u}^t(\cdot) := U_u((\bar{F}_k^t)_{k \in [K]}, \cdot)$

---
**Algorithm 5:** UBIID algorithm

---
1 **Input:** time horizon $T$, number of adversary $N$.
2 **Output:** bids for each time step $\left(\mathbf{b}^1, \mathbf{b}^2, \ldots, \mathbf{b}^{T-1}, \mathbf{b}^T\right) \in B^T$. **for** $t = 1, 2, \ldots, T$ **do**
3 $\quad\lfloor$ Play $\mathbf{b}^t := \arg\max_{\mathbf{b} \in B} \bar{u}^t(\mathbf{b})$

---

**Theorem 5.** *When facing i.i.d. adversaries in the uniform auction with bandit feedback, Algorithm 5 guarantees the regret is upper bounded by $\tilde{\mathcal{O}}\left(\sqrt{T}\right)$.*

*Proof sketch.* The previous result is essentially proved as the bound for the full-information feedback, by noticing that for all $x \in [0,1], t_{k^\star(x)} \geq t/K$. This ensures that the estimates $\bar{F}$ concentrate as well as in the full-information feedback up to additional factors which only depedn on $K$, techniques similar to the ones for Theorem 1 then yield the result. $\square$

**Lower bounds** Because the opposing bids are order statistics, previous lower bounds may not apply. Intuitively, this is because constraining the opposing bids to be ordered i.i.d. can make learning to bid easier (as Theorem 5 shows). Let us consider the discriminatory auction first. Notice that for 1-unit auction, the general setting and the i.i.d. setting coincide. Therfore the lower bound form Lemma 3 applies and regret is at least $\Omega\left(T^{2/3}\right)$. The uniform auction requires a new lower bound, that we provide below:

**Lemma 6.** *When facing i.i.d. adversaries in the uniform auction with bandit feedback, any learning algorithm must incur at least a regret of $\Omega(\sqrt{T})$.*

*Proof idea.* Focusing on two-item auctions, we build upon the proof of the lower bound of Brânzei et al., 2023 for the uniform auction in the full information feedback. Since they use opposing bids which can only be worth $0$ or $2/3$, it is possible to build distributions that mimic the behavior of the ones used in their lower bound, even when using i.i.d. bids. $\square$

**Remark 2.** *We assume knowledge of the number of adversaries $N$ in this section. However, since the components of $\boldsymbol{\beta}$ are order statistics, $N$ can also be estimated from a finite number of samples.*

## 6 Conclusion

We characterized the difficulty of learning to bid in multi-unit auctions through achievable regret rates. We show that, under full-information feedback, both auction formats can be learned with matching regret rates. In the bandit feedback setting, the picture is more nuanced: while worst-case regret guarantees are similar for both formats, the richer feedback structure of the uniform auction can make it easier to learn in instances that do not correspond to the worst-case distribution.

A key simplification in our analysis is the stochasticity assumption for opposing bids. It remains an open question whether similar results hold when adversaries are strategic, which would also enable the study of social welfare when multiple bidders learn simultaneously. Another interesting direction is to consider settings where bidders' valuations must also be learned, and to investigate how auction transparency (Cesa-Bianchi et al., 2024) impacts the learning process.

## Acknowledgements

Vianney Perchet acknowledges the support of ANR through the PEPR IA FOUNDRY project (ANR-23-PEIA-0003) and the Doom project (ANR-23-CE23-0002), as well as the ERC through the Ocean project (ERC-2022-SYG-OCEAN-101071601).

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

## Appendix

This appendix provides additional technical details, proofs, and supporting results for the main paper. It is organized as follows:

- **Section A** presents general simplifications and key properties of the auction models that are useful for our analysis.

- **Section B** contains detailed proofs of the regret upper bounds for both the full-information and bandit feedback settings, including concentration inequalities and technical lemmas.

- **Section C** provides the proofs of the main regret lower bounds for the discriminatory (first price) and uniform price auctions.

- **Section D** covers proofs and results specific to certain instances, such as $\Delta$-separated distributions and the i.i.d. unit-demand adversary setting.

Throughout, all notation, definitions, and assumptions are consistent with those in the main text, unless explicitly stated otherwise. Where appropriate, we restate key lemmas and theorems for clarity. Cross-references to the main text and within the appendix are provided for ease of navigation.

## A    Appendix

### A.1    General simplifications

This section presents general properties and simplifications of the auction models considered in the main paper. We highlight structural results that are useful for the design and analysis of learning algorithms, such as truthfulness in uniform price auctions and the implications for optimal bidding strategies. These results serve as foundational tools for the regret analyses in subsequent sections.

**Uniform price auction with unit demand**    We focus our attention on the unit demand uniform price auction, in which the bidder only wishes to acquire one item. Formally, this means that $v_1 > 0$ and for all $k > 1$, $v_k = 0$. One can notice that in this specific case, the optimal bid is straightforwardly $\mathbf{b}^* = (v_1, 0, \ldots, 0)$. Therefore, in this case, there is no need for an online learning algorithm and a bidder playing $\mathbf{b}^*$ for every time $t \in [T]$ incurs regret 0.

The following lemma shows a more general property of which the previous claim is a direct consequence.

**Lemma 7.** *In the uniform price auction, for any distribution of opposing bid $\mathcal{D}$, let $\mathbf{b}^* = (b_1^*, b_2^*, \ldots, b_K^*) := \mathrm{argmax}_{\mathbf{b} \in B} \mathop{\mathbb{E}}\limits_{\boldsymbol{\beta} \sim \mathcal{D}} [u(\mathbf{b}, \boldsymbol{\beta})]$ then $\tilde{\mathbf{b}}^\star := (v_1, \min(b_2^\star, v_2), \ldots, \min(b_K^\star, v_K))$ is also a maximizer.*

*Proof.* We will provide a stronger result, which is that, given any opposing bids $\boldsymbol{\beta} \in B$, when we denote $u(\mathbf{b}^\star, \boldsymbol{\beta})$ the utility of the bidder against these bids when bidding $\mathbf{b}^\star$, then $u(\mathbf{b}^\star, \boldsymbol{\beta}) \leq u(\tilde{\mathbf{b}}^\star, \boldsymbol{\beta})$. Let $k^\star$ be the biggest index such that $b_{k^\star}^\star > v_{k^\star}$. Denote $\tilde{\mathbf{b}}_k^\star$ the bid with $b_{k^\star}^\star$ replaced by $v_{k^\star}$. Let us consider $x(\mathbf{b}^\star, \boldsymbol{\beta})$ and $x(\tilde{\mathbf{b}}_k^\star, \boldsymbol{\beta})$, either they are equals or $x(\mathbf{b}^\star, \boldsymbol{\beta}) > x(\tilde{\mathbf{b}}_k^\star, \boldsymbol{\beta})$.

- In the first case, noticing that the price function $p(\cdot, \boldsymbol{\beta})$ is increasing in all the components of the bids, yields $u(\mathbf{b}^\star, \boldsymbol{\beta}) \leq u(\tilde{\mathbf{b}}_k^\star, \boldsymbol{\beta})$.

- In the second case, we have $x(\mathbf{b}^\star, \boldsymbol{\beta}) > x(\tilde{\mathbf{b}}_k^\star, \boldsymbol{\beta})$, which necessarily implies that $b_{k^\star}^\star \geq p(\mathbf{b}^\star, \boldsymbol{\beta}) \geq v_{k^\star}$. Therefore, $v_{k^\star} - p(\mathbf{b}^\star, \boldsymbol{\beta}) \leq 0$, noticing that this is the last term of the utility when playing $\mathbf{b}^\star$ and that the price is increasing in all the components of the bids, we can conclude that $u(\mathbf{b}^\star, \boldsymbol{\beta}) \leq u(\tilde{\mathbf{b}}_k^\star, \boldsymbol{\beta})$.

We can repeat this process for all the components of the bids. Combining this with the fact that the utility is only increasing in the first component of the bid vector (usually denoted $b_1$) allows us to get that $u(\mathbf{b}^\star, \boldsymbol{\beta}) \leq u(\tilde{\mathbf{b}}^\star, \boldsymbol{\beta})$. $\qquad\square$

Because of the previous property, when analyzing learning algorithms for the uniform price auction, we will always consider the algorithms' pick bids such that $b_1 = v_1$ when maximizing estimates of expected utility.

We now move on to providing the proofs and formulas necessary to justify our approach. We recall the Lemma 1.

**Lemma 1.** *In both auction formats, the expected utility can be expressed as a function of the bidder's bid vector* $\mathbf{b}$ *and the marginal cumulative distribution functions* $(F_k)_{k \in [K]}$.

In fact, for the discriminatory auction, the utility can be written as follows:

$$\mathop{\mathbb{E}}_{\boldsymbol{\beta} \sim \mathcal{D}} [u(\mathbf{b}, \boldsymbol{\beta})] = \sum_{i=1}^{K} F_{K-i+1}(b_i)(v_i - b_i) \tag{9}$$

In the uniform price auction, the expected utility can be written as follows :

$$\mathop{\mathbb{E}}_{\boldsymbol{\beta} \sim \mathcal{D}} [u(\mathbf{b}, \boldsymbol{\beta})] = \sum_{i=1}^{K} (F_{K-i}(b_{i+1}) - F_{K-i+1}(b_{i+1})) \left( \sum_{l=1}^{i} v_l - b_{i+1} \right) \tag{10}$$

$$+ \sum_{i=1}^{K} \mathop{\mathbb{E}}_{\beta_{K-i+1} \sim \mathcal{D}_{K-i+1}} \left[ \mathbb{1} \{ b_i \geq \beta_{K-i+1} \geq b_{i+1} \} \left( \sum_{l=1}^{i} v_l - \beta_{K-i+1} \right) \right] \tag{11}$$

*Proof.* We begin by proving the lemma for the discriminatory auction. The utility can be rewritten as $u(\mathbf{b}, \boldsymbol{\beta}) = \sum_{i=1}^{K} \mathbb{1} \{ b_i \geq \beta_{K-i+1} \} (v_i - b_i)$. Taking the expectation and rewriting by using the definition $F_k$ leads to the following equation, which concludes the proof.

$$\mathop{\mathbb{E}}_{\boldsymbol{\beta} \sim \mathcal{D}} [u(\mathbf{b}, \boldsymbol{\beta})] = \sum_{i=1}^{K} F_{K-i+1}(b_i)(v_i - b_i) \tag{12}$$

We now move to the uniform price auction. To have a convenient formula, we write the utility of the bidder in the uniform price auction by decomposing it depending on the $K + 1^{\text{th}}$ bid.

$$u(\mathbf{b}, \boldsymbol{\beta}) = \sum_{i=1}^{K} \mathbb{1} \{ \beta_{K-i} > b_{i+1} \geq \beta_{K-i+1} \} \left( \sum_{l=1}^{i} v_l - b_{i+1} \right)$$

$$+ \sum_{i=1}^{K} \mathbb{1} \{ b_i \geq \beta_{K-i+1} > b_{i+1} \} \left( \sum_{l=1}^{i} v_l - \beta_{K-i+1} \right)$$

When taking the expectation, this leads to

$$\mathop{\mathbb{E}}_{\boldsymbol{\beta} \sim \mathcal{D}} [u(\mathbf{b}, \boldsymbol{\beta})] = \sum_{i=1}^{K} \mathbb{P} (\beta_{K-i} > b_{i+1} \geq \beta_{K-i+1}) \left( \sum_{l=1}^{i} v_l - b_{i+1} \right)$$

$$+ \sum_{i=1}^{K} \mathbb{P} (b_i \geq \beta_{K-i+1} > b_{i+1}) \left( \sum_{l=1}^{i} v_l \right) - i \mathbb{E} [\mathbb{1} \{ b_i \geq \beta_{K-i+1} > b_{i+1} \} \beta_{K-i+1}]$$

Which we can simplify as

$$\mathop{\mathbb{E}}_{\boldsymbol{\beta} \sim \mathcal{D}} [u(\mathbf{b}, \boldsymbol{\beta})] = \sum_{i=1}^{K} (F_{K-i+1}(b_{i+1}) - F_{K-i}(b_{i+1})) \left( \sum_{l=1}^{i} v_l - b_{i+1} \right)$$

$$+ \sum_{i=1}^{K} (F_{K-i+1}(b_i) - F_{K-i+1}(b_{i+1})) \left( \sum_{l=1}^{i} v_l \right) \tag{13}$$

$$- \sum_{i=1}^{K} i \left( b_i F_{K-i+1}(b_i) - b_{i+1} F_{K-i+1}(b_{i+1}) - \int_{b_{i+1}}^{b_i} F_{K-i+1}(t) dt \right)$$

This concludes the proof. $\qquad\square$

With the previous proof in mind, we can define $U_u$ and $U_d$, the corresponding functions that, given the CDFs, allow us to compute the expected utility.

Let $(G_k)_{k \in [K]}$ be a sequence of CDF functions. The following equations define the functions mentioned in Lemma 1 for: **the discriminatory auction** for $\mathbf{b} \in B$:

$$U_d((G_k)_{k \in [K]}, \mathbf{b}) = \sum_{i=1}^{K} G_{K-i+1}^t (b_i) (v_i - b_i), \tag{14}$$

**the uniform auction** for $\mathbf{b} \in B$:

$$
\begin{aligned}
U_u((G_k)_{k \in [K]}, \mathbf{b}) := & \sum_{i=1}^{K} (G_{K-i+1}(b_{i+1}) - G_{K-i}(b_{i+1})) \left( \sum_{l=1}^{i} v_l - b_{i+1} \right) \\
& + \sum_{i=1}^{K} (G_{K-i+1}(b_i) - G_{K-i+1}(b_{i+1})) \sum_{l=1}^{i} v_l - i \int_{(b_{i+1}, b_i]} x dG_{K-i+1}
\end{aligned}
\tag{15}
$$

**Lemma 2.** *At time $t \in [T]$, let $\mathbf{b}^t, \boldsymbol{\beta}^t \in B$ be the bids of the learner and the adversary, in the uniform auction with bandit feedback the bidder observes $\left( \mathbb{1} \left\{ \beta_{K-i+1}^t \in (b_{i+1}^t, b_i^t] \right\} \beta_{K-i+1}^t \right)_{i \in [K]}$.*

*Proof.* At time $t$, let $i \in [K]$, then $\beta_{K-i+1}^t \in (b_{i+1}^t, b_i^t]$, is equivalent to $b_{i+1}^t < \beta_{K-i}^t \leq b_i^t$ which is in turn equivalent to $\{x(\mathbf{b}^t, \boldsymbol{\beta}^t) = i\} \cap \{p(\mathbf{b}^t, \boldsymbol{\beta}^t) \neq b_{i+1}^t\}$, which is an observable event. Note that when the event is realized, the value of $\beta_{K-i+1}^t$ is observed since it is the price.

Therefore, since it is the case for all $i \in [K]$, the bidder has observed $\left( \mathbb{1} \left\{ \beta_{K-i+1}^t \in [b_{i+1}^t, b_i^t] \right\} \beta_{K-i+1}^t \right)_{i \in [K]}$. $\square$

**Remark 3.** *Notice that the previous Lemma 2 implies that at all times $t$ and for all $i \in [K]$, the following function is also observed $\forall x \in (b_{i+1}^t, b_i^t], \mathbb{1} \left\{ \beta_{K-i}^t \in [0, x] \right\}$.*

# B Regret Upper Bounds

In this section, we provide detailed proofs of the regret upper bounds stated in the main text for both the full-information and bandit feedback settings. We include all necessary technical lemmas, concentration inequalities, and step-by-step arguments for the algorithms analyzed. Our goal is to make the derivations transparent and self-contained, enabling readers to verify each step of the analysis.

## B.1 Full-information feedback

Recall that in the full information feedback, the bidder observes $\boldsymbol{\beta}^t$ at the end of each time-step. therefore, at time $t$, it has observed $\boldsymbol{\beta}^1, \ldots, \boldsymbol{\beta}^{t-1}$.

The following lemma, which links the quality of the estimates of the marginal CDF to the quality of the estimates of the utility, can be used to derive regret guarantees for both auction formats.

9

**Lemma 8.** *Given $(\hat{F}_k^t)_{k \in [K]}$, the empirircal estimates of the cumulative marginal distributions of $\mathcal{D}$ from $t$ samples, let $\alpha \in [0, 1]$, then in both the uniform price auction (with $\hat{u}(\cdot) := U_u((\hat{F}_k)_{k \in [K]}, \cdot))$ and in the discriminatory auction (with $\hat{u}(\cdot) := U_u((\hat{F}_k)_{k \in [K]}, \cdot))$. The following inequality holds :*

$$\forall k \in [K], \mathbb{P} \left( \sup_{\mathbf{b} \in B} |\hat{u}(\mathbf{b}) - \mathbb{E}[u(\mathbf{b}, \boldsymbol{\beta})]| \leq K \epsilon^t(\alpha) \right) \geq 1 - K\alpha \tag{16}$$

*Where for the discriminatory auction $\epsilon^t(\alpha) := \sqrt{\frac{\ln\left(\frac{2}{\alpha}\right)}{2t}}$ and for the uniform auction $\epsilon^t(\alpha) := 3\sqrt{C \frac{2K \log(2K) + \log\left(\frac{1}{\alpha}\right)}{t}}$*

*Proof.* We begin with the proof of the guarantees for the discriminatory auction, which is more concise: Let $T \in \mathbb{N}$, at time $t \in [T]$, we apply the result from Massart, 1990, which in our notations provides that, for $\alpha \in [0, 1]$:

$$\forall k \in [K], \mathbb{P}\left(\sup_{x \in [0,1]} |\hat{F}_k^t(x) - F_k(x)| \leq \sqrt{\frac{\ln\left(\frac{2}{\alpha}\right)}{2t}}\right) \geq 1 - \alpha \tag{17}$$

Doing a union over the opposite events, one easily gets :

$$\mathbb{P}\left(\forall k \in [K], \sup_{x \in [0,1]} \left|\hat{F}_k^t(x) - F_k(x)\right| \leq \sqrt{\frac{\ln\left(\frac{2}{\alpha}\right)}{2t}}\right) \geq 1 - K\alpha \tag{18}$$

We can write the expected utility in the discriminatory price auction, for $\mathbf{b} = (b_1, \ldots, b_K) \in B$ as $\mathbb{E}[u(\mathbf{b}, \boldsymbol{\beta})] = \sum_{k \in [K]} F_k(b_k)(v_k - b_k)$. Using the expression of $\hat{u}$ , we get:

$$\sup_{\mathbf{b} \in B} |\hat{u}(\mathbf{b}) - \mathbb{E}[u(\mathbf{b}, \boldsymbol{\beta})]| \leq \sup_{\mathbf{b} \in B} \sum_{k \in [K]} \left|\hat{F}_k(b_k) - F_k(b_k)\right|$$

$$\leq \sum_{k \in [K]} \sup_{b_k \in [0,1]} \left|\hat{F}_k(b_k) - F_k(b_k)\right|$$

Using (18) then yields the desired result.

We now proceed with the proof for the uniform price auction. We start by rewriting the utility from (13) as follows :

$$
\begin{aligned}
\mathbb{E}_{\boldsymbol{\beta} \sim \mathcal{D}}[u(\mathbf{b}, \boldsymbol{\beta})] = {}& \mathbb{E}_{\boldsymbol{\beta} \sim \mathcal{D}}\left[\sum_{i=1}^{K} \mathbb{1}\{\beta_{K-i} > b_{i+1} \geq \beta_{K-i+1}\}\left(\sum_{j=1}^{i} v_j - b_{i+1}\right)\right.\\
&+ \left.\sum_{i=1}^{K} \mathbb{1}\{b_i \geq \beta_{K-i+1} > b_{i+1}\}\left(\sum_{j=1}^{i} v_j\right)\right]\\
&- \sum_{i=1}^{K} i\left(b_i F_{K-i+1}(b_i) - b_{i+1} F_{K-i+1}(b_{i+1}) - \int_{b_{i+1}}^{b_i} F_{K-i+1}(t)dt\right)\\
= {}& \mathbb{E}_{\boldsymbol{\beta} \sim \mathcal{D}}\left[\sum_{i=1}^{K} \mathbb{1}\{\beta_{K-i} > b_{i+1} \geq \beta_{K-i+1}\}\left(\sum_{j=1}^{i} v_j - b_{i+1}\right)\right.\\
&+ \left.\sum_{i=1}^{K} \mathbb{1}\{b_i \geq \beta_{K-i+1} > b_{i+1}\}\left(\sum_{j=1}^{i} v_j - b_{i+1}\right)\right]\\
&- \sum_{i=1}^{K} i\left(b_i - b_{i+1}\right) F_{K-i+1}(b_i)\\
&- \sum_{i=1}^{K} i \int_{b_{i+1}}^{b_i} F_{K-i+1}(t)dt
\end{aligned}
$$

The estimated utility can also be written similarly, yielding:

$$\hat{u}^t(\mathbf{b}) = \hat{\mathbb{E}}^t \left[ \sum_{i=1}^{K} \mathbb{1}\left\{ \beta_{K-i} > b_{i+1} \geq \beta_{K-i+1} \right\} \left( \sum_{j=1}^{i} v_j - b_{i+1} \right) \right.$$
$$\left. + \sum_{i=1}^{K} \mathbb{1}\left\{ b_i \geq \beta_{K-i+1} > b_{i+1} \right\} \left( \sum_{j=1}^{i} v_j - b_{i+1} \right) \right]$$
$$- \sum_{i=1}^{K} i\left( b_i - b_{i+1} \right) \hat{F}_{K-i+1}(b_i)$$
$$- \sum_{i=1}^{K} i \int_{b_{i+1}}^{b_i} \hat{F}_{K-i+1}(t) dt$$

Where $\hat{\mathbb{E}}^t$ denotes the empirical expectation.

We will show that the estimated utility is close to the true expectation by showing that it is the case for each of the terms of the sum. We begin with the terms that only require DKW bound, which we restate here: Let $T \in \mathbb{N}$, at time $t \in [T]$, the result from Massart, 1990, provides that, for $\alpha \in [0, 1]$:

$$\forall k \in [K], \mathbb{P}\left( \sup_{x \in [0,1]} |\hat{F}_k^t(x) - F_k(x)| \leq \sqrt{\frac{\ln\left(\frac{2}{\alpha}\right)}{2t}} \right) \geq 1 - \alpha \qquad (19)$$

We can therefore get that with probability $1 - K\alpha$:

$$\left| \sum_{i=1}^{K} i \int_{b_{i+1}}^{b_i} F_{K-i+1}(t) dt - \sum_{i=1}^{K} i \int_{b_{i+1}}^{b_i} \hat{F}_{K-i+1}(t) dt \right| \leq \sum_{i=1}^{K} i(b_i - b_{i+1})\epsilon \leq K\epsilon \qquad (20)$$

Similarly, with probability $1 - K\alpha$:

$$\left| \sum_{i=1}^{K} i\left( b_i - b_{i+1} \right) F_{K-i+1}(b_i) - \sum_{i=1}^{K} i\left( b_i - b_{i+1} \right) \hat{F}_{K-i+1}(b_i) \right| \leq \sum_{i=1}^{K} i(b_i - b_{i+1})\epsilon \leq K\epsilon \qquad (21)$$

Note that (20) and (21) are also valid when taking the supremum over $b \in B$.

We now move to bounding the last difference needed. We first provide the required tools.

Usual concentration inequality for the CDF can be obtained by considering the family of indicator functions : $x \mapsto \mathbb{1}\{X \leq x\}$. In order to obtain tighter concentration bounds, we will use results on uniform convergence results on multiclass learnability from Shalev-Shwartz and Ben-David, 2014 to bound all the terms simultaneously.

First notice that for any $\boldsymbol{\beta} \in B$, the family of events $(\{\beta_{K-i} > b_{i+1} \geq \beta_{K-i+1}\})_{i \in [K]} \cup (\{b_i \geq \beta_{K-i+1} > b_{i+1}\})_{i \in [K]}$ are disjoint for any $\mathbf{b} \in B$. We will consider the multiclass classification functions associated with these events for $\mathbf{b} \in B$, and denote $\mathcal{H}_B$ this family of classification functions. We also denote $f$ the function which maps an element of $B$ to the corresponding multiclassifier.

We begin by bounding the Natarajan dimension of $\mathcal{H}_B$, a generalisation of VC dimension. For a reminder of the definition, see Shalev-Shwartz and Ben-David, 2014 section **29.1**.

Notice that amongst the events, there are interval indicator functions for each coordinate of the vector $\boldsymbol{\beta}$. Therefore, we can use the same argument as for obtaining the VC dimension of the family of functions $\mathbb{1}\{X \in [a, b]\}$. The main difference being that we require $2K + 2$ element of $B$ in order to have at least three different values amongst one coordinate, than observing that one cannot obtain labels $[1, 0, 1]$ in the corresponding class yields the following upper bound on the Natarajan dimension $\mathcal{N}$ of the family of multiclassifier at hand:

$$\mathcal{N}\left(\mathcal{H}_B\right) \leq 2K + 2 \qquad (22)$$

Then, using the Multiclass Fundamental Theorem Shalev-Shwartz and Ben-David, 2014, **Theorem 29.3** yields the following concentration inequality, with probability $1 - \alpha$ :

$$\sup_{\mathbf{b} \in B} \left| \frac{\sum_{t=1}^{T} f(\mathbf{b})(\boldsymbol{\beta}^t)}{T} - \mathbb{E}_{\boldsymbol{\beta} \sim \mathcal{D}} [f(\mathbf{b})(\boldsymbol{\beta})] \right| \leq \epsilon^t \tag{23}$$

Where there is a universal constant $C > 0$ such that $\epsilon^T := \sqrt{C \frac{2K \log(K) + \log\left(\frac{1}{\alpha}\right)}{T}}$

We can therefore use this bound in order to obtain the following with probability at least $1 - \alpha$:

$$\sup_{\mathbf{b} \in B} \left| \hat{\mathbb{E}}^t \left[ \sum_{i=1}^{K} \left( \mathbb{1} \left\{ \beta_{K-i} > b_{i+1} \geq \beta_{K-i+1} \right\} + \mathbb{1} \left\{ b_i \geq \beta_{K-i+1} > b_{i+1} \right\} \right) \left( \sum_{j=1}^{i} v_j - b_{i+1} \right) \right] \right.$$
$$\left. - \hat{\mathbb{E}}^t \left[ \sum_{i=1}^{K} \left( \mathbb{1} \left\{ \beta_{K-i} > b_{i+1} \geq \beta_{K-i+1} \right\} + \mathbb{1} \left\{ b_i \geq \beta_{K-i+1} > b_{i+1} \right\} \right) \left( \sum_{j=1}^{i} v_j - b_{i+1} \right) \right] \right|$$
$$\leq K \epsilon^t \tag{24}$$

With this equation, we can now bound the error of the expected utility using (20), (21), and (24), with probability at least $1 - \alpha$:

$$\sup_{\mathbf{b} \in B} \left| \hat{u}^t(\mathbf{b}) - \mathbb{E}_{\boldsymbol{\beta} \sim \mathcal{D}} [u(\mathbf{b}, \boldsymbol{\beta})] \right| \leq 3K \epsilon^t \tag{25}$$

which concludes the proof. □

We restate the theorem providing upper bounds on the regret in the full-information feedback :

**Theorem 1.** *Under full-information feedback, Algorithm 1 achieves a regret* $R_T = \tilde{\mathcal{O}} \left( K \sqrt{T} \right)$ *for both the discriminatory and the uniform price auction.*

*Proof.* Lemma 8 yields the following concentration inequality :

$$\mathbb{P} \left( \left| \hat{u}^t(\mathbf{b}) - \mathbb{E}_{\boldsymbol{\beta} \sim \mathcal{D}} [u(\mathbf{b}, \boldsymbol{\beta})] \right| \leq K \epsilon^t \right) \geq 1 - K \alpha \tag{26}$$

Let's denote the optimal bid $\mathbf{b}^\star := \operatorname{argmax}_{\mathbf{b} \in B} \mathbb{E}_{\boldsymbol{\beta} \sim \mathcal{D}} [u(\mathbf{b}, \boldsymbol{\beta})]$. Since $\mathbf{b}^t$ is the maximum of $\hat{u}^t$, we known that $\hat{u}^t(\mathbf{b}^\star) \leq \hat{u}^t(\mathbf{b}^t)$. Applying the previous concentration inequality to both $\mathbf{b}^\star$ and $\mathbf{b}^t$ and leveraging the optimality of $\mathbf{b}^t$ gives us with probability $1 - K \alpha$:

$$\mathbb{E}_{\boldsymbol{\beta} \sim \mathcal{D}} \left[ u(\mathbf{b}^t, \boldsymbol{\beta}) \right] \geq \mathbb{E}_{\boldsymbol{\beta} \sim \mathcal{D}} \left[ u(\mathbf{b}^\star, \boldsymbol{\beta}) \right] + 6K \epsilon^t \tag{27}$$

Choosing $\alpha = \frac{1}{T^2}$ and summing over $t \in [T]$, then yields the desired regret bounds : $R_T = \tilde{\mathcal{O}} \left( K \sqrt{T} \right)$.

□

## B.2 Bandits Feedback

We provide in this section the details necessary in order to obtain the guarantees for the algorithms 2 and 3. This subsection is therefore only focused on the uniform price auction, which has the most expressive bandit feedback of the two auction formats.

Let us recall that in this setting, the bidder only observes the outcome of the auctions at the end of each time step. Formally at time $t$, with $\boldsymbol{\beta}^t$ the bid of the adversary and $\mathbf{b}^t$ its own bid, the bidder observes $x(\boldsymbol{\beta}^t, \mathbf{b}^t)$ and $p(\boldsymbol{\beta}^t, \mathbf{b}^t)$. Lemma 2 provides an alternative representation, while Remark 3 points out how it relates to indicator functions used for CDFs empirical estimates.

We recall the formula of the estimates of the marginal CDFs for $k \in [K]$ :

$$\forall x \in [0, 1], \tilde{F}_k(x) := \frac{\sum_{j=1}^{t} \mathbb{1} \left\{ x \in (b_{K+2-k}^t, b_{K+1-k}^t] \right\} \mathbb{1} \left\{ \beta_k^t \leq x \right\}}{\sum_{j=1}^{t} \mathbb{1} \left\{ x \in (b_{K+2-k}^t, b_{K+1-k}^t] \right\}} \tag{28}$$

### B.2.1 Explore then commit algorithm

We restate the theorem providing upper bounds on the regret in the bandit setting for the uniform auction.

**Theorem 2.** *The estimate then commit algorithms, with the exploration time* $T_{expl} = K^{2/3}T^{2/3}$ *achieves a regret bound of* $\tilde{\mathcal{O}}\left(K^{5/3}T^{2/3}\right)$

*Proof.* In order to prove the regret of the algorithm, we provide a two-part bound which matches the two-phase behavior of the algorithm. Indeed, we use the following regret decomposition between exploration regret and commitment regret :

$$R_T = R_T^{expl} + R_T^{com} \tag{29}$$

Where we define the exploration regret as

$$R_T^{expl} = T_{expl} \sup_{\mathbf{b} \in B} \left( \mathbb{E}_{\boldsymbol{\beta} \sim \mathcal{D}}[u(\mathbf{b}, \boldsymbol{\beta})] \right) - \sum_{t=1}^{T_{expl}} \mathbb{E}_{\boldsymbol{\beta}^t \sim \mathcal{D}}\left[u(\mathbf{b}^t, \boldsymbol{\beta}^t)\right], \tag{30}$$

and the commitment regret as

$$R_T^{com} = (T - T_{expl}) \sup_{\mathbf{b} \in B} \left( \mathbb{E}_{\boldsymbol{\beta} \sim \mathcal{D}}[u(\mathbf{b}, \boldsymbol{\beta})] \right) - \sum_{t=T_{expl}}^{T} \mathbb{E}_{\boldsymbol{\beta}^t \sim \mathcal{D}}\left[u(\mathbf{b}^t, \boldsymbol{\beta}^t)\right]. \tag{31}$$

We begin by upper-bounding the exploration regret by noticing that the utility is upper-bounded by $K$. This leads to

$$R_T^{expl} \leq K T_{expl} \tag{32}$$

Notice that during the exploration phase, the bidder ensures that for every $k \in [K]$ it has at least observed $\lfloor T_{expl}/K \rfloor$ samples of $\beta_k$.

Notice that during the exploration phase, for every $k \in [K]$, the bidder has observed at least $\lfloor T_{expl}/K \rfloor$ samples of $\beta_k$. This follows directly from applying Lemma 2 to the bids in the exploration phase. This ensures our CDFS estimates $\tilde{F}$ are valid on the whole domain and that we can create concentration bands.

We will use concentration results on the empirical CDF to show that our estimate of the expected utility is close to the actual one. We begin by stating the concentration results that we will use :

We can use Massart, 1990 to show that for all $k \in [K]$: :

$$\forall k \in [K], \mathbb{P}\left( \sup_{x \in [0,1]} \left| \tilde{F}_k^{T_{expl}}(x) - F_k(x) \right| \leq \sqrt{\frac{\ln\left(\frac{2}{\alpha}\right)}{2\lfloor T_{expl}/K \rfloor}} \right) \geq 1 - \alpha \tag{33}$$

We will need another concentration result. It is well known that the family of indicator functions of intervals has VC dimensions 2, therefore, it follows from section 12.5 of Boucheron et al., 2013 that for any $k \in [K]$:

$$\mathbb{P}\left( \sup_{(a,b)^2 \in [0,1], a<b} \left| \frac{\tilde{F}_k^{T_{expl}}(b) - \tilde{F}_k^{T_{expl}}(a) - F_k(b) + F_k(a)}{\sqrt{F_k(b) - F_k(a)}} \right| \leq \epsilon\left(\lfloor T_{expl}/K \rfloor \alpha\right) \right) \geq 1 - \alpha \tag{34}$$

Where $\epsilon(T, \alpha) = \sqrt{2 \frac{\ln\left(\frac{2}{\alpha}\right) + \log(eT)}{T}}$.

To upper bound $\sup_{b \in B} \left| \mathbb{E}_{\boldsymbol{\beta} \sim \mathcal{D}}(\mathbf{b}, \boldsymbol{\beta}) - \tilde{u}^{T_{expl}}(\mathbf{b}) \right|$, we will use the following formulation of the expected utility into terms well suited for showing uniform convergence.

$$\mathbb{E}\left[u(\mathbf{b},\boldsymbol{\beta})\right] = \sum_{i=1}^{K} F_{K-i+1}(b_i)v_i - \sum_{i=1}^{K} ib_{i+1}\left(F_{k-i+1}(b_{i+1}) - F_{k-i}(b_{i+1})\right)$$

$$- \sum_{i=1}^{K} ib_{i+1}\left(F_{k-i+1}(b_i) - F_{K-i+1}(b_{i+1})\right) \tag{35}$$

$$- \sum_{i=1}^{K} i\int_{b_{i+1}}^{b_i} F_{K-i+1}(t)dt - \sum_{i=1}^{K} i\left(b_i - b_{i+1}\right)F_{K-i+1}(b_i)$$

Notice first that we can obtain the two following equations as in the proof of Lemma 8, we can use (33) to obtain similar bounds as in (20) and (21).

$$\left|\sum_{i=1}^{K} i\int_{b_{i+1}}^{b_i} F_{K-i+1}(t)dt - \sum_{i=1}^{K} i\int_{b_{i+1}}^{b_i} \tilde{F}_{K-i+1}^{T_{expl}}(t)dt\right| \leq \sum_{i=1}^{K} i(b_i - b_{i+1})\epsilon \leq K\epsilon \tag{36}$$

$$\left|\sum_{i=1}^{K} i\left(b_i - b_{i+1}\right)F_{K-i+1}(b_i) - \sum_{i=1}^{K} i\left(b_i - b_{i+1}\right)\tilde{F}_{K-i+1}^{T_{expl}}(b_i)\right| \leq \sum_{i=1}^{K} i(b_i - b_{i+1})\epsilon \leq K\epsilon \tag{37}$$

We can also get the following bound:

$$\left|\sum_{i=1}^{K} \left(F_{K-i+1}(b_i) - \tilde{F}_{K-i+1}^{T_{expl}}(b_i)\right)v_i\right| \leq K\epsilon \tag{38}$$

It only remains to get concentration bounds for the following term:

$$\left|\sum_{i=1}^{K} ib_{i+1}\left(2F_{k-i+1}(b_{i+1}) - F_{k-i}(b_{i+1}) - F_{K-i+1}(b_i)\right)\right.$$

$$\left. - ib_{i+1}\left(2\tilde{F}_{k-i+1}(b_{i+1}) - \tilde{F}_{k-i}(b_{i+1}) - \tilde{F}_{K-i+1}(b_i)\right)\right| \tag{39}$$

We start by the following computations :

$$\sum_{i=1}^{K} ib_{i+1}\left(F_{K-i+1}(b_{i+1}) - F_{K-i}(b_{i+1})\right) = \sum_{i=1}^{K} ib_{i+1}\left(F_{K-i+1}(b_{i+1})\right)$$

$$- \sum_{i=2}^{K+1} (i-1)b_i\left(F_{K-i+1}(b_i)\right)$$

$$= \sum_{i=1}^{K} b_{i+1}\left(F_{K-i+1}(b_{i+1})\right) + \sum_{i=2}^{K} b_{i+1}(i-1)\left(F_{K-i+1}(b_{i+1}) - F_{K-i+1}(b_i)\right)$$

$$- \sum_{i=2}^{K+1} (i-1)(b_i - b_{i+1})\left(F_{K-i+1}(b_i)\right) - Kb_{K+1}$$

Adding the other terms in $F$ of (39) yields :

$$\sum_{i=1}^{K} ib_{i+1}\left(2F_{k-i+1}(b_{i+1}) - F_{k-i}(b_{i+1}) - F_{K-i+1}(b_i)\right) = \sum_{i=1}^{K} b_{i+1}\left(F_{K-i+1}(b_{i+1})\right)$$

$$+ \sum_{i=2}^{K} b_{i+1}(2i-1)\left(F_{K-i+1}(b_{i+1}) - F_{K-i+1}(b_i)\right)$$

$$- \sum_{i=2}^{K+1} (i-1)(b_i - b_{i+1})\left(F_{K-i+1}(b_i)\right) - Kb_{K+1}$$

$$\tag{40}$$

Naturally, $\tilde{u}$ can be written the same way, we will therefore bound (39) by bounding the error on each sum independently. By upper bounding using both (33) and (34) we can upper bound the last terms as follows, with probability $1 - 2K\alpha$ :

$$
\sup_{\mathbf{b} \in B} \left| \sum_{i=1}^{K} ib_{i+1} \left( 2F_{k-i+1}(b_{i+1}) - F_{k-i}(b_{i+1}) - F_{K-i+1}(b_i) \right) \right.
$$
$$
\left. -ib_{i+1} \left( 2\tilde{F}_{k-i+1}(b_{i+1}) - \tilde{F}_{k-i}(b_{i+1}) - \tilde{F}_{K-i+1}(b_i) \right) \right| \leq K\epsilon + K^{3/2}\epsilon + K\epsilon
$$

(41)

Combining equations therefore yields with probability $1 - 2K\alpha$

$$
\sup_{b \in B} \left| \mathbb{E}_{\boldsymbol{\beta} \sim \mathcal{D}}(\mathbf{b}, \boldsymbol{\beta}) - \tilde{u}^{T_{expl}}(\mathbf{b}) \right| \leq 6K^{3/2}\epsilon
$$

(42)

Where $\epsilon = \sqrt{2 \frac{\ln\left(\frac{2}{\alpha}\right) + \log(e\lfloor T_{expl}/K \rfloor)}{\lfloor T_{expl}/K \rfloor}}$.

Then notice from Algorithm 2 that for all $t > T_{expl}$, all the bids $\mathbf{b}^t$ are the same (because they are the solution of the same maximization problem). By optimality we get $\tilde{u}^{T_{expl}}(\mathbf{b}^{T_{expl}}) \geq \tilde{u}^{T_{expl}}(\mathbf{b}^\star)$. Combining with (42) then yields for all $t \geq T_{expl}$ with probability $1 - 2K\alpha$:

$$
\mathbb{E}\left[u(\mathbf{b}^\star, \boldsymbol{\beta})\right] - \mathbb{E}\left[u(\mathbf{b}^t, \boldsymbol{\beta})\right] \leq 6K^{3/2}\epsilon
$$

(43)

By summing over $T_{expl} \leq t \leq T$ and recognizing the formula of $R_T^{com}$ we get, with probability $1 - 2K\alpha$:

$$
R_T^{com} \leq 12K^{5/3}T^{2/3} \sqrt{\ln\left(\frac{2}{\alpha}\right) + \ln(e\lfloor T_{expl}/K \rfloor)}.
$$

(44)

Since $T_{expl} = K^{2/3}T^{2/3}$.

And from (32) we get:

$$
R_T^{expl} \leq K^{5/3}T^{2/3}
$$

(45)

Choosing $\alpha = \frac{1}{T^2}$ and summing $R_T^{com}$ and $R_T^{expl}$ yields the desired regret bounds.

$\square$

### B.2.2 Improved bandit algorithm

We now move to the improved bandit algorithm, which can allow for a better regret bound when the instance it faces makes it possible.

We restate the theorem providing upper bounds on the regret in the bandit setting, including the instance-dependent bounds. Before providing the proof, we introduce several instance-dependent quantities which are relevant for the instance-dependent bounds and the proofs.

**Theorem 4.** *Algorithm 3 guarantee a regret of $\tilde{\mathcal{O}}(K^{5/3}T^{2/3})$ in general and, when $\Delta > 0$, an instance-dependent regret bounds of $\tilde{\mathcal{O}}(K\sqrt{T})$.*

*Proof.* For the analysis, we will consider two cases, one that we call the worst-case, and one that is the good instance. We talk about good instances if during the first phase of the algorithm for some $t^\star$, $\Delta^{t^\star}$ is positive, and if not, it is a worst-case instance.

**Worst-case** Let us first analyse what happens in worst-case instances : $\forall t \in [T_{expl}], \Delta^t \leq 0$. As for the proof of Theorem 2, we decompose the regret into exploration and rest regret.

$$
R_T = R_T^{expl} + R_T^{com}
$$

(46)

We can, by noticing the utility is positive and smaller than $K$, upper bound the exploration regret by $KT_{expl}$, which yields

$$
R_T^{expl} \leq K^{5/3}T^{2/3}
$$

(47)

In order to provide a similar analysis of the regret of the second phase as for Algorithm 2, we need to show that after the exploration phase, the bidder has effectively observed enough samples of each component of $\boldsymbol{\beta} \sim \mathcal{D}$ on the right intervals (because observation is no longer guaranteed on $[0, 1]$).

We recall the formula of the marginal CDF estimators for $k \in [K]$:

$$\forall x \in [0, 1], \tilde{F}_k(x) := \frac{\sum_{j=1}^{t} \mathbb{1}\left\{x \in (b_{K+2-k}^t, b_{K+1-k}^t]\right\} \mathbb{1}\left\{\beta_k^t \le x\right\}}{\sum_{j=1}^{t} \mathbb{1}\left\{x \in (b_{K+2-k}^t, b_{K+1-k}^t]\right\}} \tag{48}$$

It suffices to notice that during the exploration phase, at time $t$, when $k_t := t - K\lfloor t/K \rfloor + 1$, the interval corresponding to $\tilde{F}_{K-k_t+1}$, that reads $\mathbb{1}\left\{x \in (b_{k_t+1}^t, b_{k_t}^t]\right\}$ strictly contains $\mathcal{I}_{k_t}^t \cup \mathcal{I}_{k_t+1}^t$, and by monotonicity of the intervals, contains $\mathcal{I}_k^{t'} \cup \mathcal{I}_{k+1}^{t'}$ for any $t' \ge t$. We can therefore write : For any $t \in [T]$, for any $k \in [K]$, $\forall x \in \mathcal{I}_k^t \cup \mathcal{I}_{k+1}^t$, $\sum_{j=1}^{t} \mathbb{1}\left\{x \in (b_{k_t+1}^t, b_{k_t}^t]\right\} \ge \min(\lfloor t/K \rfloor, \lfloor T_{expl}/K \rfloor)$.

Notice from (15) that for $t' \ge t$, for $\mathbf{b} \in \prod_{k \in [K]} \mathcal{I}_k^{t'}$, $\tilde{F}_{K-k_t+1}$ is only evaluated on $\mathcal{I}_k^{t'} \cup \mathcal{I}_{k+1}^{t'}$. The previous statement is therefore sufficient for the concentration results of Theorem 2 to be reproduced.

The only modification needed is to notice that we can only guarantee that the optimal bid $\mathbf{b}^\star$ remains in the $\prod_{k \in [K]} \mathcal{I}_k^{t'}$ with probability $1 - K\alpha$ at each time. The probability of the bad event "the best bids are out of our confidence intervals" is at most $K\alpha T$

We can therefore obtain that with probability $1 - 2K\alpha T$ :

$$R_T^{com} \le 12K^{5/3}T^{2/3}\sqrt{\ln\left(\frac{2}{\alpha}\right) + \ln(e\lfloor T_{expl}/K \rfloor)}. \tag{49}$$

Choosing $\alpha = \frac{1}{T^3}$ and summing $R_T^{com}$ and $R_T^{expl}$ yields the desired worst-case regret bounds of $K^{5/3}T^{2/3}$.

**Instance dependent**    In this section of the proofs, we examine the case where for some $t^* \in [T_{expl}]$, we have $\Delta^{t^*} > 0$. First note that because intervals are *shrinking*, this implies that for all $t \ge t^*$, $\Delta^t > 0$.

As for the previous case, we focus our attention on how samples that allow us to estimate the marginal CDFs are collected. The same proof as in the previous section holds to show that, for times $t \le t^*$, at least $\lfloor t/K \rfloor$ samples of each $\beta_{K-k+1}$ are observed, on the relevent intervals: $\mathcal{I}_k^t \cup \mathcal{I}_{k+1}^t$.
We now focus on the behavior of the samples in the *else* statement, i.e. when $t \ge t^\star$. Let $t \in [T]$ such that $\Delta^t > 0$ and denote $k_t' := t - 2K\lfloor t/2K \rfloor + 2$ and $k_t := \lfloor k_t'/2 \rfloor$. Then notice that if $k_t'$ is even $\mathcal{I}_{k_t}^t \subseteq (b_{k_t'+1}^t, b_{k_t'+1}^t]$ and if $k_t'$ is odd $\mathcal{I}_{k_t+1}^t \subseteq (b_{k_t'+1}^t, b_{k_t'+1}^t]$. Therefore, after one round (i.e. $2K$ steps) the bidder observes at least one sample of $\beta_{K-k+1}$ on $\mathcal{I}_k^t \cup \mathcal{I}_{k+1}^t$.

We can therefore write the two following concentration inequalities as in the proof of Algorithm 2, here on specific intervals, and in any time variant. For any $t \in [T]$ and $k \in [K]$ :

The first from Massart, 1990:

$$\forall k \in [K], \mathbb{P}\left(\sup_{x \in \mathcal{I}_k^t \cup \mathcal{I}_{k+1}^t} \left|\tilde{F}_{K-k+1}^t(x) - F_{K-k+1}(x)\right| \le \sqrt{\frac{\ln\left(\frac{2}{\alpha}\right)}{2\lfloor t/2K \rfloor}}\right) \ge 1 - \alpha \tag{50}$$

The second :

$$\mathbb{P}\left(\sup_{(a,b)^2 \in \mathcal{I}_k^t \cup \mathcal{I}_{k+1}^t} \left|\frac{\tilde{F}_{K-k+1}^t(b) - \tilde{F}_{K-k+1}^t(a) - F_{K-k+1}(b) + F_{K-k+1}(a)}{\sqrt{F_k(b) - F_k(a)}}\right| \le \epsilon\left(\lfloor t/2K \rfloor\alpha\right)\right) \ge 1 - \alpha \tag{51}$$

Where $\epsilon(t, \alpha) = \sqrt{2\frac{\ln\left(\frac{2}{\alpha}\right) + \log(et)}{t}}$.

This leads, using the same procedure as in Theorem 2 to with probability $1 - 2K\alpha$

$$\sup_{\mathbf{b} \in \prod_{i \in [K]} \mathcal{I}_k^t} \left|\mathbb{E}_{\boldsymbol{\beta} \sim \mathcal{D}}[u(\mathbf{b}, \boldsymbol{\beta})] - \tilde{u}^t(\mathbf{b})\right| \le 6K^{3/2}\epsilon \tag{52}$$

Where $\epsilon = \sqrt{2 \frac{\ln\left(\frac{2}{\alpha}\right) + \log(e \lfloor T_{expl}/K \rfloor)}{\lfloor T_{expl}/K \rfloor}}$.

Using the optimality of $u_{max}^t$, and the fact that as long as (52) is true, $\mathbf{b}^\star$ remains in $\prod_{i \in [K]} \mathcal{I}_k^t$ ensures that, for $t \geq t'$ with proba $1 - tK\alpha$ :

$$\mathbb{E}_{\boldsymbol{\beta} \sim \mathcal{D}} \left[ u(\mathbf{b}^t, \boldsymbol{\beta}) \right] \geq \mathbb{E}_{\boldsymbol{\beta} \sim \mathcal{D}} [u(\mathbf{b}^\star, \boldsymbol{\beta})] + 12K^{3/2} \epsilon^t \tag{53}$$

The cumulative regret therefore follows by summing and choosing $\alpha = \frac{1}{T^3}$.

$$R_T \leq Kt^* + \sum_{t=t^*}^{T} 12K^{3/2} \epsilon^t \leq Kt^* + \sum_{t=t^*}^{T} 12K^2 \sqrt{\frac{12 \log t}{t}} \tag{54}$$

All that is left is to show that we can upper bound $t^*$ based on some instance-dependent quantity. We introduce these quantities here: we denote $B^\star \subset B$ the set of maximizers of the expected utility and the optimal intervals $\mathcal{I}_k^\star \subset \mathbf{R}$ the smallest closed intervals such that $B^\star \subset \prod_{k=1}^K \mathcal{I}_k^\star$. We also introduce the interval gap $\Delta := \min_{k \in \{2,\dots,K\}} \left( \min \mathcal{I}_k^\star - \max \mathcal{I}_{k+1}^\star \right)$, and the utility gaps: $\delta_k^+ := \mathbb{E}[u(\mathbf{b}^\star, \boldsymbol{\beta})] - \max_{\mathbf{b} s.t. b_k \leq \min b_k^\star - \Delta/2} \mathbb{E}[u((\mathbf{b}), \boldsymbol{\beta})]$. and $\delta_k^- := \mathbb{E}[u(\mathbf{b}^\star, \boldsymbol{\beta})] - \max_{\mathbf{b} s.t. b_k \geq \max b_k^\star + \Delta/2} \mathbb{E}[u((\mathbf{b}), \boldsymbol{\beta})]$.

Notice from equation (52), that necessarily, as soon as $12K^{3/2} \epsilon^t \leq \min((\delta_k +, \delta_k^-)_{k \in [K]})$, for all $k \in [K]$, the intervals $\mathcal{I}_k^t$ are at most $\Delta/2$ wider than $\mathcal{I}_k^\star$, and therefore $\Delta^t > 0$. Therefore, $t^* \leq \frac{12K}{\min((\delta_k +, \delta_k^-)_{k \in [K]})}^2$.

With these quantities, we can now write the following regret bound, which completes the proof :

$$R_T \leq \tilde{\mathcal{O}} \left( \min \left( K^{5/3} T^{2/3}, \frac{12K}{\min((\delta_k +, \delta_k^-)_{k \in [K]}, \max(\Delta, 0))}^2 + K^2 \sqrt{T} \right) \right) \tag{55}$$

$\square$

# C   Regret lower bouds

This section contains the proofs of the main regret lower bounds for the discriminatory (first price) and uniform price auctions, namely Lemma 3 and Theorem 3. We construct hard instances and use information-theoretic arguments to establish fundamental limits on the achievable regret in these settings. The techniques used here are inspired by and extend prior work, as referenced in the main text. The *difficult instance* that we consider is significantly more straightforward to describe for the proof of Lemma 3. We therefore begin with this one, as it should make understanding both proofs easier.

## C.1   Lower bound for the first price auction

Let us restate the lemma before providing the proof.

**Lemma 3.** *Any algorithms for bidding in repeated first price auctions with known valuation must incur a regret of $\Omega\left(T^{2/3}\right)$*

*Proof.* The first-price auction is a special case of the discriminatory auction when only one item is available. The utility simply writes: $u(b, \beta) = \mathbb{1}\{b \geq \beta\}(v - b)$. Notably, when $\beta$ has a random distribution of Cumulative Distribution Function $F$, $\mathbb{E}[u(b, \beta)] = F(b)(v - b)$.

Throughout this proof, we will use these notations, with the bid of the bidder $b \in [0, 1]$ and the opposing bid $\beta \in [0, 1]$.

To prove a lower bound on the regret of learning in the repeated first price auction, we use techniques similar to those in Cesa-Bianchi et al., 2024 (ie, we show that in this problem we can embed $T^{1/3}$ "bandit arms" into our continuous actions space).

**Outline of the proof** We devise a base probability distribution of $\beta$, the adversary's bid, which is described by its CDF $F$. This distribution is such that the utility of the bidder is constant on a wide range of bids (ie, $[0, c]$ with $c$ a constant). From this base probability, we derive a family of $(\tilde{F}_i)$, multiple slightly perturbed versions of $F$. The main characteristic of the perturbation is that it leaves the CDF of $F$ unchanged outside of an interval of width $\sim T^{-1/3}$. We provide $\tilde{\mathcal{O}}\left(T^{1/3}\right)$ of such perturbation. The instance we consider is a bidder such that the distribution of the opposing bids is drawn uniformly at random amongst the perturbed distribution. Quantifying the distance between the feedback distribution generated by each perturbed distribution allows us to prove the lower bound.

The base distribution $\mathcal{D}$ is characterized by its CDF $F$ that we define to be:

$$F(b) = \frac{1}{3(1-b)} \mathbb{1}\left\{b < \frac{1}{3}\right\} + \left(\frac{1}{4} + \frac{3b}{4}\right) \mathbb{1}\left\{b \geq \frac{1}{3}\right\} \tag{56}$$

It is easy to check that $\mathbb{E}_{\beta \sim \mathcal{D}}\left[u(\cdot, \beta)\right]$ is constant on $[0, \frac{1}{3}]$, and is decreasing on $[\frac{1}{3}, 1]$.

We define the intervals $\left(\mathcal{I}_i^T\right)_{(i \in [0, \lfloor 3T^{1/3} \rfloor])}$ on which the $i^{\text{th}}$ perturbation takes place as follows :
$\mathcal{I}_i^T := \left[\frac{i}{9}T^{-1/3}, \frac{i+1}{9}T^{-1/3}\right)$.

Let's then define the corresponding perturbations $\varepsilon_i^T : [0, 1] \longrightarrow \mathbb{R}$:

$$\varepsilon_i^T(b) = \left(F\left((i+1)T^{-1/3}\right) - F(b)\right) \mathbb{1}\left\{b \in [i \in \mathcal{I}_i^T]\right\} \tag{57}$$

We can now define $F_i^T := F + \varepsilon_i^T$, the $i^{\text{th}}$ perturbed CDF, and $\mathcal{D}_i^T$ the corresponding distribution. It is straightforward to see that $F_i^T$ and $F$ coincide outside of $\mathcal{I}_i^T$. It is also clear, since $F_i^T$ is constant on $\mathcal{I}_i^T$, that the utility is maximized in $\frac{i}{9}T^{-1/3}$.

We can now describe the instance faced by the bidder : Let $T \in \mathbb{N}$, the difficult instance is as follows:

- An index $i \in \left[\lfloor 3T^{1/3} \rfloor\right]$ is drawn uniformly at random.

- For t in $[T]$:

    - The bidder picks a bid $b^t$ based on the past feedback $Z_1, ..., Z_{t-1}$

    - The adversaries bid $\beta^t$ is drawn according to $\mathcal{D}_i^T$, the distribution with CDF $F_i^T$.

    - The bidder receives the feedback $Z_t = \mathbb{1}\{b^t \geq \beta^t\}$, and the utility $u_t := (v - b^t)\mathbb{1}\{b^t \geq \beta^t\}$

From here onward, we focus on determining the minimum regret any algorithm must incur when facing the previously described instance of the first-price auction. We also restrict our analysis to *sensible* algorithms, that is, whose bids are always in $[0, \frac{1}{2}]$. These assumptions are relaxed at the end of the proof.

**Some Notations:** We denote $\mathbb{P}^0$ the probability measure over $\beta$ when its drawn according to $\mathcal{D}$ and $\mathbb{P}^i$ the probability measure of $\beta$ conditionally on the value of $i$ (ie when its drawn according to $\mathcal{D}_i^T$). The notation extends naturally to expectations $\mathbb{E}^0$ and $\mathbb{E}^i$ as well as regret $R_T^0$ and $R_T^i$.

For $t \in [T]$, we denote $Y_t : [0, 1] \longrightarrow \{0, 1\}$ such that $Y_t(b) = \mathbb{1}\{b \geq \beta_t\}$. And $Z_t$ is the feedback received at time $t$, it is a random variable and depends on the bid played by the algorithm $\mathcal{A}$ at time $t$. For now, we restrict ourselves to deterministic algorithms, which allows us to write $b^t : \{0, 1\}^{t-1} \longrightarrow [0, 1]$ the function which maps the feedback received up to time $t - 1$ to the bid played by the algorithm $\mathcal{A}$ at time $t$. With these notation, notice that $Z_t = Y_t(b^t(Z_1, ..., Z_{t-1}))$.

With these in mind, the following lemma constitutes the core of the proof. We postpone its proofs just after the end of this proof, essentially, one uses the fact that distributions of feedbacks $Z_t$ are too close to be distinguished.

**Lemma 9.** *For all $i \leq \lceil 3T^{-1/3} \rceil$ and any deterministic algorithm $\mathcal{A}$, denoting $N_T(i)$ the number of times algorithm $\mathcal{A}$ produces a bid in $\mathcal{I}_i^T$, we have :*

$$\mathbb{E}^i\left[N_T(i)\right] - \mathbb{E}^0\left[N_T(i)\right] \leq \frac{9}{40}T^{2/3}\sqrt{\mathbb{E}^0[N_T(i)]} \tag{58}$$

We now use the fact that we know that in the hard instance, $i$ is drawn uniformly at random.

Averaging for $i \leq \lceil 3T^{1/3} \rceil$, we get:

$$\frac{\sum_{i=1}^{\lceil 3T^{1/3} \rceil} \mathbb{E}^i \left[ N_T(i) \right]}{\lceil 3T^{1/3} \rceil} \leq \frac{\sum_{i=1}^{\lceil 3T^{1/3} \rceil} \mathbb{E}^0 \left[ N_T(i) \right]}{\lceil 3T^{1/3} \rceil} + \frac{9}{40} T^{2/3} \sqrt{\frac{\sum_{i=1}^{\lceil 3T^{1/3} \rceil} \mathbb{E}^0 [N_T(i)]}{\lceil 3T^{1/3} \rceil}} \tag{59}$$

$$\leq \frac{T}{\lceil 3T^{1/3} \rceil} + \frac{9}{40} T^{2/3} \sqrt{\frac{T}{\lceil 3T^{1/3} \rceil}} \tag{60}$$

$$R_T = \frac{1}{\lceil 3T^{1/3} \rceil} \sum_{i=1}^{\lceil 3T^{1/3} \rceil} r_i \mathbb{E}^i \left[ T - N_T(i) \right] \tag{61}$$

$$\geq \min_i r_i \left( T - \frac{T}{\lceil 3T^{1/3} \rceil} - \frac{9}{40} T^{2/3} \sqrt{\frac{T}{\lceil 3T^{1/3} \rceil}} \right) \tag{62}$$

$$\geq \min_i r_i \left( T - \frac{T}{3T^{1/3}} - \frac{9}{40} T^{2/3} \sqrt{\frac{T}{3T^{1/3}}} \right) \tag{63}$$

$$\geq \min_i r_i \left( T - \frac{1}{3} T^{2/3} - \frac{3\sqrt{3}}{40} T \right) \tag{64}$$

Where $r_i$ is the sub-optimality of not playing interval $i$ when it is optimal.

$$r_i = \frac{1}{3 \left( 1 - \frac{i+1}{9} T^{-1/3} \right)} \left( 1 - \frac{i}{9} T^{-1/3} \right) - \frac{1}{3}$$

$$r_i = \frac{1}{9} T^{-1/3} \frac{1}{3 \left( 1 - \frac{i+1}{9} T^{-1/3} \right)} \geq \frac{1}{27} T^{-1/3}$$

Reinjecting this lower bound on $r_i$ yields the desired lower bound of $\Omega \left( T^{2/3} \right)$. This bound is valid for any algorithm which issues bids in $[0, \frac{1}{2}]$, Lemma 11 generalises it to any deterministic bid. The lower bound generalizes to random algorithms by Yao's minimax principle.

$\square$

### C.1.1 Technical lemma

**Lemma 9.** *For all $i \leq \lceil 3T^{-1/3} \rceil$ and any deterministic algorithm $\mathcal{A}$, denoting $N_T(i)$ the number of times algorithm $\mathcal{A}$ produces a bid in $\mathcal{I}_i^T$, we have :*

$$\mathbb{E}^i \left[ N_T(i) \right] - \mathbb{E}^0 \left[ N_T(i) \right] \leq \frac{9}{40} T^{2/3} \sqrt{\mathbb{E}^0 [N_T(i)]} \tag{58}$$

*Proof.* We want to upper bound the following quantity :

$$\mathbb{E}^i \left[ N_T(i) \right] - \mathbb{E}^0 \left[ N_T(i) \right] = \sum_{t=1}^{T} \mathbb{P}^i \left[ b^t \left( Z_1, .., Z_{t-1} \right) \in \mathcal{I}_i^T \right] - \mathbb{P}^0 \left[ b^t \left( Z_1, .., Z_{t-1} \right) \in \mathcal{I}_i^T \right] \tag{65}$$

We will upper each term of the sum independently. The algorithm can be anything deterministic; therefore, $b^t(.)$ can be any function, which is why we bound the previous quantity as follows :

$$\mathbb{P}^i\left[b^t\left(Z_1,..,Z_{t-1}\right)\in\mathcal{I}_i^T\right]-\mathbb{P}^0\left[b^t\left(Z_1,..,Z_{t-1}\right)\in\mathcal{I}_i^T\right]$$

$$\tag{66}$$

$$\leq\left\|\mathbb{P}^i_{(Z_1,..,Z_{t-1})}-\mathbb{P}^0_{(Z_1,..,Z_{t-1})}\right\|_{TV}\tag{67}$$

$$\leq\sqrt{\frac{1}{2}KL\left(\mathbb{P}^i_{(Z_1,..,Z_{t-1})};\mathbb{P}^0_{(Z_1,..,Z_{t-1})}\right)}\tag{68}$$

$$\leq\sqrt{\frac{1}{2}\left(KL\left(\mathbb{P}^i_{Z_1};\mathbb{P}^0_{Z_1}\right)+\sum_{j=2}^t KL\left(\mathbb{P}^i_{Z_j|Z_1,..,Z_{j-1}};\mathbb{P}^0_{Z_j|Z_1,..,Z_{j-1}}\right)\right)}\tag{69}$$

Where $\mathbb{P}^i_{(Z_1,..,Z_{t-1})}$ is the push-forward probability of $(Z_1,..,Z_{t-1})$.

Since we focus on deterministic algorithms, $b^j(Z_1,..,Z_{j-1})$ conditionally on $Z_1,..,Z_{t-1}$ is fixed.

$$KL\left(\mathbb{P}^i_{Z_j|Z_1,..,Z_{j-1}};\mathbb{P}^0_{Z_j|Z_1,..,Z_{j-1}}\right)=\mathbb{E}^0\left[\ln\left(\frac{\mathbb{P}^0\left[Z_j=0\mid Z_1,..,Z_{j-1}\right]}{\mathbb{P}^i\left[Z_j=0\mid Z_1,..,Z_{j-1}\right]}\right)\mathbb{P}^0\left[Z_j=0\mid Z_1,..,Z_{j-1}\right]\right.$$

$$\tag{70}$$

$$+\ln\left(\frac{\mathbb{P}^0\left[Z_j=1\mid Z_1,..,Z_{j-1}\right]}{\mathbb{P}^i\left[Z_j=1\mid Z_1,..,Z_{j-1}\right]}\right)\mathbb{P}^0\left[Z_j=1\mid Z_1,..,Z_{j-1}\right]\right]\tag{71}$$

$$=\mathbb{E}^0\left[\left(\ln\left(\frac{\mathbb{P}^0\left[Y_j(b^j)=0\right]}{\mathbb{P}^i\left[Y_j(b^j)=0\right]}\right)\mathbb{P}^0\left[Y_j(b^j)=0\right]\right.\right.\tag{72}$$

$$\left.\left.+\ln\left(\frac{\mathbb{P}^0\left[Y_j(b^j)=1\right]}{\mathbb{P}^i\left[Y_j(b^j)=1\right]}\right)\mathbb{P}^0\left[Y_j(b^j)=1\right]\right)\mathbb{1}\left\{b^j\in\mathcal{I}_i^T\right\}\right]\tag{73}$$

$$\leq\frac{\left(F^0(b^j)-F^i(b^j)\right)^2}{F^i(b^j)\left(1-F^i(b^j)\right)}\mathbb{P}^0\left(b^j\in\mathcal{I}_i^T\right)\tag{74}$$

Where $\mathbb{1}\left\{b^j\in\mathcal{I}_i^j\right\}$ appears because for other values of $b^j$, the logarithmic terms are 0.

We obtain (74) from the fact that the $KL$ between two Bernoulli $\mathcal{P},\mathcal{Q}$ of parameters $p,q$ is $KL(\mathcal{P},\mathcal{Q})\leq\frac{(p-q)^2}{q(1-q)}$.

Since $F^i$ is non-decreasing and $F^i(b^t)\in[\frac{1}{3},\frac{1}{2}]$, by concavity of $x\mapsto x(1-x)$ on this interval.

$$KL\left(\mathbb{P}^i_{Z_j|Z_1,..,Z_{j-1}};\mathbb{P}^0_{Z_j|Z_1,..,Z_{j-1}}\right)\leq\frac{\left(F^0(b^t)-F^i(b^t)\right)^2}{\frac{2}{9}}\mathbb{P}^0\left(b^t\in\mathcal{I}_i^T\right)\tag{75}$$

$$\leq\frac{9}{2}\left(F^0\left(\frac{i}{9}T^{-1/3}\right)-F^0\left(\frac{i+1}{9}T^{-1/3}\right)\right)^2\mathbb{P}^0\left(b^t\in\mathcal{I}_i^T\right)\tag{76}$$

We can then use Lemma 10 in combination with (69) and (76) to obtain:

$$\mathbb{P}^i\left[b^t\left(Z_1,..,Z_{t-1}\right)\in\mathcal{I}_i^T\right]-\mathbb{P}^0\left[b^t\left(Z_1,..,Z_{t-1}\right)\in\mathcal{I}_i^T\right]\leq\frac{9}{40}T^{-1/3}\sqrt{\mathbb{E}^0[N_t(i)]}\tag{77}$$

We can therefore re-inject in (65) to get :

$$\mathbb{E}^i\left[N_T(i)\right]-\mathbb{E}^0\left[N_T(i)\right]\leq\sum_{t=1}^T\frac{9}{40}T^{-1/3}\sqrt{\mathbb{E}^0[N_t(i)]}\leq\frac{9}{40}T^{2/3}\sqrt{\mathbb{E}^0[N_T(i)]}\tag{78}$$

$\square$

**Lemma 10.** *For $i \leq \lceil 3T^{1/3} \rceil$, we have the following inequality :*

$$\left( F^0(\frac{i}{9}T^{-1/3}) - F^0(\frac{i+1}{9}T^{-1/3}) \right)^2 \leq \frac{9T^{-2/3}}{400} \tag{79}$$

*Proof.*

$$\left( F^0\left(\frac{i}{9}T^{-1/3}\right) - F^0\left(\frac{i+1}{9}T^{-1/3}\right) \right)^2 = \left( \frac{1}{3\left(1 - \frac{i}{9}T^{-1/3}\right)} - \frac{1}{3\left(1 - \frac{i+1}{9}T^{-1/3}\right)} \right)^2 \tag{80}$$

$$= \left( 3\frac{1}{9 - iT^{-1/3}} - \frac{1}{9 - (i+1)T^{-1/3}} \right)^2 \tag{81}$$

$$= \left( \frac{3T^{-1/3}}{\left(9 - iT^{-1/3}\right)\left(9 - (i+1)T^{-1/3}\right)} \right)^2 \tag{82}$$

$$\leq \frac{9T^{-2/3}}{400} \tag{83}$$

Where we get (83) from the fact that $i \leq \lceil 3T^{1/3} \rceil \implies \left(9 - iT^{-1/3}\right)\left(9 - (i+1)T^{-1/3}\right) \geq 20$, (given $T > 1$ but thats always the case) $\square$

**Lemma 11.** *Let $\mathcal{A}$ be any algorithm for the first price auction, their exists an algorithm $\mathcal{A}^*$ achieving a lower regret that only plays bids in $[0, \frac{1}{2}]$.*

*Proof.* For any algorithm $\mathcal{A}$, if at time $t$ the algorithms picks a bid $b^t$ outside of $[0, \frac{1}{2}]$, replace $b^t$ by $\tilde{b}^t = \frac{1}{4}$, ignore the feedback received from the auction and provide a simulated feedback drawn according to $(F(b^t))$ to $\mathcal{A}$. Because the feedback we provide follows the same distribution, and both are drawn independently from any other randomness (namely other $\beta^t$), on expectation, the algorithm behaves identically for the following time steps, and utility-wise the cumulative reward receive by the algorithms has strictly increased, since $u(\tilde{b}^t) \geq u(b^t)$.

$\square$

### C.2 Lower bound for the uniform price auction

This section is dedicated to showing the lower bound on the rate of learning in uniform price auctions. The lower bounds is valid when the learner is willing to acquire at least 3 items (ie $v_3 > 0$, and $K \geq 3$).

We restate the theorem :

**Theorem 3.** *In the uniform price auction, any algorithms must incur a regret of $\Omega\left(T^{2/3}\right)$ for learning to bid with known valuations, when valuations are such that $v_3 > 0$.*

*Proof.* For simplicity, we prove the lower bounds in the uniform 3-item auction. The lower bound naturally extends to $K > 3$ by considering adversary bid distribution where for all $k \leq K - 3$, $\mathbb{P}(\beta_k = 1) = 1$. The proof revolves around constructing a hard instance of the learning to bid problem, and then showing one cannot avoid regret, which grows as $T^{2/3}$ in this instance.

**The hard instance** In the instance of the problem, we focus on the bidder who has the following valuations $v_1, v_2, v_3 = 1, \frac{1}{3}, \frac{1}{3}$. As before, $\boldsymbol{\beta} = (\beta_1, \beta_2, \beta_3)$ is the adversary's bid and for $k \in \{1, 2, 3\}$, $F_k$ is the CDF of $\beta_k$, we also denote $f_k$ the corresponding density. Since the quantity that matters in our problems are the marginals CDFs (Lemma 1), we first define our hard instance by the densities and marginals (checking then that there indeed exists a distribution with these is then enough).

The overview of the hard instance is as follows: we characterize (via the marginals) a distribution $\mathcal{D}$ (called base instance) of adversary bids such that there exists an interval $I \subset [0, 1]$ such that the set of maximizers of the expected utility can be written as $v_1, y, y$ for $y \in I$. We then build a family of $\tilde{\mathcal{O}}(T^{1/3})$ distributions based on perturbed versions of $\mathcal{D}$ (via additive perturbation of the marginals)

for which there exists a single expected utility maximizer. The hard instance then consists in selecting uniformly at random one of the perturbed distributions when $t = 0$.

The densities of the base instance are as follows:

$$
f_3(y) = \begin{cases} 101 & \text{if } y \in [0, \frac{5}{900}] \\ \frac{1}{3(v_3 - y)} & \text{if } y \in (\frac{5}{900}, \frac{1}{6}] \\ 0 & \text{if } y \in (\frac{1}{6}, \frac{1}{3}] \\ \alpha_3 & \text{if } y \in (\frac{1}{3}, 1] \end{cases}
$$

$$
f_2(y) = \begin{cases} 21 & \text{if } y \in [0, \frac{5}{900}] \\ \frac{1}{3(v_2 - y)} & \text{if } y \in (\frac{5}{900}, \frac{1}{6}] \\ 0 & \text{if } y \in (\frac{1}{6}, \frac{1}{3}] \\ \alpha_2 & \text{if } y \in (\frac{1}{3}, 1] \end{cases}
$$

$$
f_1(y) = \begin{cases} 1 & \text{if } y \in [0, \frac{5}{900}] \\ \frac{1}{3(v_2 - y)} & \text{if } y \in (\frac{5}{900}, \frac{1}{6}] \\ 0 & \text{if } y \in (\frac{1}{6}, \frac{1}{3}] \\ \alpha_1 & \text{if } y \in (\frac{1}{3}, 1] \end{cases}
$$

Where $\alpha_1, \alpha_2, \alpha_3$ are normalisation constant. *The values of $\boldsymbol{\alpha}$ are not important because they define the density above $v_2$ and $v_3$, ie in regions of the bid space $B$ that we know are suboptimal from Lemma 7.*

This results in the following CDFs for $y \in [0, v_2]$ :

$$
F_3(y) = \begin{cases} 101y & \text{if } y \in [0, \frac{5}{900}] \\ \frac{5}{9} + F_1(y) & \text{if } y \in (\frac{5}{900}, \frac{1}{6}] \\ \frac{5}{9} + F_1(\frac{1}{6}) & \text{if } y \in (\frac{1}{6}, \frac{1}{3}] \end{cases}
$$

$$
F_2(y) = \begin{cases} 21y & \text{if } y \in [0, \frac{5}{900}] \\ \frac{1}{9} + F_1(y) & \text{if } y \in (\frac{5}{900}, \frac{1}{6}] \\ \frac{1}{9} + F_1(\frac{1}{6}) & \text{if } y \in (\frac{1}{6}, \frac{1}{3}] \end{cases}
$$

$$
F_1(y) = \begin{cases} y & \text{if } y \in [0, \frac{5}{900}] \\ \frac{1}{3}\left[\ln\left(v_2 - \frac{5}{900}\right) - \ln\left(v_2 - y\right)\right] + \frac{5}{900} & \text{if } y \in (\frac{5}{900}, \frac{1}{6}] \\ F_1(\frac{1}{6}) & \text{if } y \in (\frac{1}{6}, \frac{1}{3}] \end{cases}
$$

As before, the opponents' bids follow a distribution, $\boldsymbol{\beta} \sim \mathcal{D}$. Given a sample of this random variable $\boldsymbol{\beta} = (\beta_1, \beta_2, \beta_3)$ the utility of the learner is the following function :

$$
\begin{aligned}
u_{\boldsymbol{\beta}}(b_1, b_2, b_3) = & \mathbb{1}\left\{b_1 \geq \beta_3 \geq b_2\right\}(v_1 - \beta_3) \\
& + \mathbb{1}\left\{\beta_2 \geq b_2 \geq \beta_3\right\}(v_1 - b_2) \\
& + \mathbb{1}\left\{b_2 \geq \beta_2 \geq b_1\right\}(v_1 + v_2 - 2\beta_2) \\
& + \mathbb{1}\left\{\beta_1 \geq b_3 \geq \beta_2\right\}(v_1 + v_2 - 2b_3) \\
& + \mathbb{1}\left\{b_3 \geq \beta_1\right\}(v_1 + v_2 + v_3 - 3\beta_1)
\end{aligned}
$$

For $k \in [K]$, $F_k : [0, 1] \to [0, 1]$ is the CDF of the $k^{th}$ element of the vector $\boldsymbol{\beta}$. And we denote the corresponding density $f_k$. The expected utility then writes :

$$
\begin{aligned}
\mathbb{E}_{\boldsymbol{\beta} \sim \mathcal{D}}\left[u_{\boldsymbol{\beta}}(b_1, b_2, b_3)\right] = & v_1\left(F_3(b_1) - F_3(b_2)\right) - \int_{b_2}^{b_1} y f_3(y) dy \\
& + \left[F_3(b_2) - F_2(b_2)\right](v_1 - b_2) \\
& + (v_1 + v_2)\left(F_2(b_2) - F_2(b_1)\right) - 2\int_{b_3}^{b_2} y f_2(y) dy \\
& + \left[F_2(b_3) - F_1(b_3)\right](v_1 + v_2 - 2b_3) \\
& + (v_1 + v_2 + v_3) F_1(b_3) - 3\int_{0}^{b_3} y f_1(y) dy
\end{aligned}
$$

We can hence compute partial derivatives :

$$\frac{\partial}{\partial b_1} \mathop{\mathbb{E}}_{\boldsymbol{\beta} \sim \mathcal{D}} [u_{\boldsymbol{\beta}}(b_1, b_2, b_3)] = (v_1 - b_1) f_3(b_1) \tag{84}$$

$$\frac{\partial}{\partial b_2} \mathop{\mathbb{E}}_{\boldsymbol{\beta} \sim \mathcal{D}} [u_{\boldsymbol{\beta}}(b_1, b_2, b_3)] = F_2(b_2) - F_3(b_2) + (v_2 - b_2) f_2(b_2) \tag{85}$$

$$\frac{\partial}{\partial b_3} \mathop{\mathbb{E}}_{\boldsymbol{\beta} \sim \mathcal{D}} [u_{\boldsymbol{\beta}}(b_1, b_2, b_3)] = 2 (F_1(b_3) - F_2(b_3)) + (v_3 - b_3) f_1(b_3) \tag{86}$$

We can therefore write the partial derivative explicitly :

$$\frac{\partial}{\partial b_2} \mathop{\mathbb{E}}_{\boldsymbol{\beta} \sim \mathcal{D}} [u_{\boldsymbol{\beta}}(b_1, b_2, b_3)] = \begin{cases} \frac{21}{3} - 101y & \text{if } y \in \left[0, \frac{5}{900}\right] \\ -\frac{1}{9} & \text{if } y \in \left(\frac{5}{900}, \frac{1}{6}\right] \\ -\frac{4}{9} & \text{if } y \in \left(\frac{1}{6}, \frac{1}{3}\right] \end{cases} \tag{87}$$

$$\frac{\partial}{\partial b_3} \mathop{\mathbb{E}}_{\boldsymbol{\beta} \sim \mathcal{D}} [u_{\boldsymbol{\beta}}(b_1, b_2, b_3)] = \begin{cases} \frac{1}{3} - 21y & \text{if } y \in \left[0, \frac{5}{900}\right] \\ +\frac{1}{9} & \text{if } y \in \left(\frac{5}{900}, \frac{1}{6}\right] \\ -\frac{2}{9} & \text{if } y \in \left(\frac{1}{6}, \frac{1}{3}\right] \end{cases} \tag{88}$$

Recall, that $\mathbf{b} \in B \implies b_2 \geq b_3$, it should be clear form (88) and (87) that the maximum is attained for all $b_2, b_3 \in \left[\frac{5}{900}, \frac{1}{6}\right]$, such that $b_2 = b_3$. Furthermore, there exist $c \in \mathbb{R}$ such that for all $b_2, b_3 \in \left[\frac{5}{900}, \frac{1}{6}\right]$, if $b_2 \neq b_3$ the utility at least $c(b_2 - b_3)$ sub-optimal.

Let $\epsilon > 0$, for $k \in \left[0, \left\lfloor \frac{1}{7\epsilon} \right\rfloor\right]$ we denote $I_\epsilon^k := \left(\frac{5}{900} + k\epsilon, \frac{5}{900} + (k+1)\epsilon\right)$. We will use these intervals to define local perturbations of the CDFs. Note that with these notation, $I_{\frac{\epsilon}{2}}^{2k}$ and $I_{\frac{\epsilon}{2}}^{2k+1}$ are respectively the first and second half of $I_\epsilon^k$.

For the purpose of the lower bound, we only need to introduce a perturbation of $F_2$. We therefore define $F_{2,\epsilon}^k$ as follows :

$$F_{2,\epsilon}^k(y) = \begin{cases} F_2(y) + (y - k\epsilon) & \text{if } y \in I_{\frac{\epsilon}{2}}^{2k} \\ F_2(y) + \epsilon - (y - k\epsilon) & \text{if } y \in I_{\frac{\epsilon}{2}}^{2k+1} \\ F_2(y) & \text{otherwise} \end{cases} \tag{89}$$

With the following density

$$f_{2,\epsilon}^k(y) = \begin{cases} f_2(y) + 1 & \text{if } y \in I_{\frac{\epsilon}{2}}^{2k} \\ f_2(y) - 1 & \text{if } y \in I_{\frac{\epsilon}{2}}^{2k+1} \\ f_2(y) & \text{otherwise} \end{cases} \tag{90}$$

We therefore define $\mathcal{D}_\epsilon^k \in \mathcal{P}\left([0,1]^3\right)$, a distribution such that if $(\beta_1, \beta_2, \beta_3) \sim \mathcal{D}_\epsilon^k$ then the corresponding CDFs of $\beta_1, \beta_2, \beta_3$ are $F_1, F_{2,\epsilon}^k, F_3$.

We have now defined what we need to define the hard instance. It is defined by the following process:

- Given $T$, set $\epsilon$

- Draw $k$ uniformly at random in $\left[0, \frac{1}{7\epsilon}\right]$

- For t in $[T]$:
    - The bidder picks bids $b_1^t, b_2^t, b_3^t$ based on the past feedbacks $Z_1, ..., Z_{t-1}$.
    - The adversaries bids are drawn :$\boldsymbol{\beta}^t \sim \mathcal{D}_\epsilon^k$.
    - The bidder receives the utility $u_{\boldsymbol{\beta}^t}(b_1^t, b_2^t, b_3^t)$ and the feedback $Z_t := Z(\boldsymbol{\beta}^t, \mathbf{b}^t)$

Note that $\epsilon$ depends on $T$, intuitively, it should behave as $T^{-\frac{1}{3}}$. The exact dependency required is determined later in the proof.

Because in this instance, the only thing a bidder can learn is $F_{2,\epsilon}^k$ or equivalently which distribution $\mathcal{D}_\epsilon^k$ it faces, we will discard in the analysis the part of the feedback which doesn't provide information about the second marginal distribution. Reducing to $Z_t := Z(\boldsymbol{\beta}^t, \mathbf{b}^t) = (\mathbb{1}\{\beta_2^t \leq b_3\}, \mathbb{1}\{\beta_2^t \in (b_3, b_2]\}\beta_2^t, \mathbb{1}\{\beta_2^t > b_2\})$ (this is directly implied by Lemma 2). To provide a simpler analysis, it is convenient to work with a one dimesional feedback, notice that the previously described feedback is equivalent (one can define a bijection) with $Z(\boldsymbol{\beta}^t, \mathbf{b}^t) = \max(\mathbb{1}\{\beta_2^t \in (b_3, b_2]\}\beta_2^t, \mathbb{1}\{\beta_2^t > b_2\})$, this is the formulation we use in the following proofs. We also only consider algorithms that, at all times $t$, bid $b_1^t = v_1$. This is without loss of generality, as not doing so would only add additional regret and provide no additional useful feedback.

We also only consider deterministic algorithms, the lower bound then generalizes to randomized algorithms by Yao's minimax principle.

Our final reduction is the following: we consider in the following analysis only algorithms such that $\sum_{t=1}^T b_2^t - b_3^t = o(T^{2/3})$. This is without loss of generality: since the optimal bids $b_2^\star$ and $b_3^\star$ are always equal and belong to $I_\epsilon^k$. The expression of the derivatives (87) and (88) ensures that for all bids with $b_2$ and $b_3$ in the interval $[5/900, \frac{1}{6}]$ the algorithms incurs a regret which scales with $b_2 - b_3$. Furthermore, if either $b_2$ or $b_3$ is not in $[5/900, \frac{1}{6}]$, the algorithms incur an additional regret term of constant size. Therefore any algorithms such that we do not have $\sum_{t=1}^T b_2^t - b_3^t = o(T^{2/3})$ incurs a regret which scales as $\Omega(T^{2/3})$ in this instance.

**Some Notations:** We denote $\mathbb{P}^0$ a probability measure over $\boldsymbol{\beta}$ when the CDFs are respectively $F_1, F_2, F_3$ and $\mathbb{P}^k$ a probability measure over $\boldsymbol{\beta}$ conditionally on the value of $k$ (ie when its distributed according to $\mathcal{D}_\epsilon^k$). The notation extends naturally to expectations $\mathbb{E}^0$ and $\mathbb{E}^k$.

**Lemma 12.** *For all $i \leq \lceil 3T^{-1/3} \rceil$ and any deterministic algorithm $\mathcal{A}$, denoting $N_T(i)$ the number of times algorithm $\mathcal{A}$ produces a bid such that $(b_3^t, b_2^t] \cap \mathcal{I}_i^T \neq \emptyset$, we have :*

$$\mathbb{E}^i\left[N_T(i)\right] - \mathbb{E}^0\left[N_T(i)\right] \leq 10\epsilon T \sqrt{\mathbb{E}^0[N_T(i)]} \tag{91}$$

Averaging for $i \leq \lfloor \frac{1}{7\epsilon} \rfloor$, we get:

$$\frac{\sum_{i=1}^{\lfloor \frac{1}{7\epsilon} \rfloor} \mathbb{E}^i\left[N_T(i)\right]}{\lfloor \frac{1}{7\epsilon} \rfloor} \leq \frac{\sum_{i=1}^{\lfloor \frac{1}{7\epsilon} \rfloor} \mathbb{E}^0\left[N_T(i)\right]}{\lfloor \frac{1}{7\epsilon} \rfloor} + 10\epsilon T \sqrt{\frac{\sum_{i=1}^{\lfloor \frac{1}{7\epsilon} \rfloor} \mathbb{E}^0[N_T(i)]}{\lfloor \frac{1}{7\epsilon} \rfloor}} \tag{92}$$

$$\leq \frac{T}{\lfloor \frac{1}{7\epsilon} \rfloor} + 10\epsilon T \sqrt{\frac{T}{\lfloor \frac{1}{7\epsilon} \rfloor}} \tag{93}$$

Let us denote $\tilde{N_T}(i)$ the number of times algorithm $\mathcal{A}$ produces a bid such that $(b_3^t, b_2^t] \cap \mathcal{I}_i^T \neq \emptyset$, and $i$ is the smallest such index. We can recover for this one that $\sum_{i=1}^{\lfloor \frac{1}{7\epsilon} \rfloor} \mathbb{E}^0\left[\tilde{N}_T(i)\right] = T$. Notice that because we focus on algorithms such that $\sum_{t=1}^T b_2^t - b_3^t = o(T^{2/3})$, the difference $\sum_{i=1}^{\lfloor \frac{1}{7\epsilon} \rfloor} \mathbb{E}^0\left[\tilde{N}_T(i)\right] - \sum_{i=1}^{\lfloor \frac{1}{7\epsilon} \rfloor} \mathbb{E}^0\left[N_T(i)\right] = o\left(\frac{T^{2/3}}{\epsilon}\right)$. We continue the computation without this additional term of size $o\left(\frac{T^{2/3}}{\epsilon}\right)$ as it is clear the bound remains unchanged from this addition (it would only add to a term of order $o\left(\frac{T^{2/3}}{\epsilon}\right)$ in the right hand side of the inequality).

$$R_T \geq \frac{1}{\lfloor \frac{1}{7\epsilon} \rfloor} \sum_{i=1}^{\lfloor \frac{1}{7\epsilon} \rfloor} r_i \mathbb{E}^i\left[T - N_T(i)\right] \tag{94}$$

$$\geq \min_i r_i \left(T - \frac{T}{\lfloor \frac{1}{7\epsilon} \rfloor} - 10\epsilon T \sqrt{\frac{T}{\lfloor \frac{1}{7\epsilon} \rfloor}}\right) \tag{95}$$

Where $r_i$ is the sub-optimality of not playing bids in interval $i$ when it is optimal. Using (87) and (88) we obtain

$$r_i \geq \frac{\epsilon}{12}$$

Choosing $\epsilon = \frac{1}{700} T^{-1/3}$ yields the desired lower bound of $\Omega\left(T^{2/3}\right)$

$\square$

### C.2.1 Technical lemma

**Lemma 12.** *For all $i \leq \lceil 3T^{-1/3} \rceil$ and any deterministic algorithm $\mathcal{A}$, denoting $N_T(i)$ the number of times algorithm $\mathcal{A}$ produces a bid such that $(b_3^t, b_2^t) \cap \mathcal{I}_i^T \neq \emptyset$, we have :*

$$\mathbb{E}^i \left[ N_T(i) \right] - \mathbb{E}^0 \left[ N_T(i) \right] \leq 10 \epsilon T \sqrt{\mathbb{E}^0[N_T(i)]} \tag{91}$$

*Proof.* For convenience and to have shorter expression, we denote $(b_3^t, b_2^t](\cdot) := (b_3^t(\cdot), b_2^t(\cdot)]$.

We want to upper bound the following quantity :

$$\mathbb{E}^i \left[ N_T(i) \right] - \mathbb{E}^0 \left[ N_T(i) \right] = \sum_{t=1}^{T} \mathbb{P}^i \left[ (b_3^t, b_2^t] (Z_1, .., Z_{t-1}) \cap \mathcal{I}_i^T \neq \emptyset \right] \tag{96}$$
$$- \mathbb{P}^0 \left[ (b_3^t, b_2^t] (Z_1, .., Z_{t-1}) \cap \mathcal{I}_i^T \neq \emptyset \right]$$

We will upper each term of the sum independently. The algorithm can be anything deterministic, therefore $(b_3^t, b_2^t](.)$ can be any function, which is why we bound the previous quantity as follows :

$$\mathbb{P}^i \left[ (b_3^t, b_2^t] (Z_1, .., Z_{t-1}) \cap \mathcal{I}_i^T \neq \emptyset \right] - \mathbb{P}^0 \left[ (b_3^t, b_2^t] (Z_1, .., Z_{t-1}) \cap \mathcal{I}_i^T \neq \emptyset \right]$$

$$\tag{97}$$

$$\leq \left\| \mathbb{P}^i_{(Z_1, .., Z_{t-1})} - \mathbb{P}^0_{(Z_1, .., Z_{t-1})} \right\|_{TV} \tag{98}$$

$$\leq \sqrt{\frac{1}{2} KL \left( \mathbb{P}^i_{(Z_1, .., Z_{t-1})} ; \mathbb{P}^0_{(Z_1, .., Z_{t-1})} \right)} \tag{99}$$

$$\leq \sqrt{\frac{1}{2} \left( KL \left( \mathbb{P}^i_{Z_1} ; \mathbb{P}^0_{Z_1} \right) + \sum_{j=2}^{t} KL \left( \mathbb{P}^i_{Z_j | Z_1, .., Z_{j-1}} ; \mathbb{P}^0_{Z_j | Z_1, .., Z_{j-1}} \right) \right)} \tag{100}$$

Where $\mathbb{P}^i_{(Z_1, .., Z_{t-1})}$ is the push-forward probability of $(Z_1, .., Z_{t-1})$.

Since we focus on deterministic algorithms, $b^j(Z_1, .., Z_{j-1})$ conditionally on $Z_1, .., Z_{j-1}$ is fixed. We can therefore write the following equation to bound terms in the square root of (100).

$$KL \left( \mathbb{P}^i_{Z_j | Z_1, .., Z_{j-1}} ; \mathbb{P}^0_{Z_j | Z_1, .., Z_{j-1}} \right) = \mathbb{E}^0 \left[ \ln \left( \frac{\mathbb{P}^0 \left[ Z_j = 0 \mid Z_1, .., Z_{j-1} \right]}{\mathbb{P}^i \left[ Z_j = 0 \mid Z_1, .., Z_{j-1} \right]} \right) \mathbb{P}^0 \left[ Z_j = 0 \mid Z_1, .., Z_{j-1} \right] \right. \tag{101}$$

$$+ \int_0^1 \ln \left( \frac{p^0 \left[ Z_j = u \mid Z_1, .., Z_{j-1} \right]}{p^i \left[ Z_j = u \mid Z_1, .., Z_{j-1} \right]} \right) p^0 \left[ Z_j = u \mid Z_1, .., Z_{j-1} \right] du \tag{102}$$

$$\left. + \ln \left( \frac{\mathbb{P}^0 \left[ Z_j = 1 \mid Z_1, .., Z_{j-1} \right]}{\mathbb{P}^i \left[ Z_j = 1 \mid Z_1, .., Z_{j-1} \right]} \right) \mathbb{P}^0 \left[ Z_j = 1 \mid Z_1, .., Z_{j-1} \right] \right] \tag{103}$$

Note that this can be written as such because $Z_j$ takes values 1 and 0 with positive probability, it can also take values in $b_3^j, b_2^j$ and its distribution admits a density because the distribution of $\beta_2$ admits a

density. We continue forward in our calculation, using the fact that the distribution of $\beta_2$ and hence $Z_j$ between $\mathbb{P}^0$ and $\mathbb{P}^i$ only differs on $\mathcal{I}_\epsilon^i$.

$$KL\left(\mathbb{P}^i_{Z_j|Z_1,..,Z_{j-1}}; \mathbb{P}^0_{Z_j|Z_1,..,Z_{j-1}}\right) = \mathbb{E}^0\left[\ln\left(\frac{\mathbb{P}^0\left[Z_j=0 \mid Z_1,..,Z_{j-1}\right]}{\mathbb{P}^i\left[Z_j=0 \mid Z_1,..,Z_{j-1}\right]}\right)\mathbb{P}^0\left[Z_j=0 \mid Z_1,..,Z_{j-1}\right]\right.$$
(104)

$$+\sum_{k=0}^{\frac{1}{7\epsilon}}\int_{I_\epsilon^k}\ln\left(\frac{p^0\left[Z_j=u \mid Z_1,..,Z_{j-1}\right]}{p^i\left[Z_j=u \mid Z_1,..,Z_{j-1}\right]}\right)p^0\left[Z_j=u \mid Z_1,..,Z_{j-1}\right]du$$
(105)

$$\left.+\ln\left(\frac{\mathbb{P}^0\left[Z_j=1 \mid Z_1,..,Z_{j-1}\right]}{\mathbb{P}^i\left[Z_j=1 \mid Z_1,..,Z_{j-1}\right]}\right)\mathbb{P}^0\left[Z_j=1 \mid Z_1,..,Z_{j-1}\right]\right]$$
(106)

$$= \mathbb{E}^0\left[\ln\left(\frac{\mathbb{P}^0\left[Z_j=0 \mid Z_1,..,Z_{j-1}\right]}{\mathbb{P}^i\left[Z_j=0 \mid Z_1,..,Z_{j-1}\right]}\right)\mathbb{P}^0\left[Z_j=0 \mid Z_1,..,Z_{j-1}\right]\right.$$
(107)

$$+\int_{I_\epsilon^i}\ln\left(\frac{p^0\left[Z_j=u \mid Z_1,..,Z_{j-1}\right]}{p^i\left[Z_j=u \mid Z_1,..,Z_{j-1}\right]}\right)p^0\left[Z_j=u \mid Z_1,..,Z_{j-1}\right]du$$
(108)

$$\left.+\ln\left(\frac{\mathbb{P}^0\left[Z_j=1 \mid Z_1,..,Z_{j-1}\right]}{\mathbb{P}^i\left[Z_j=1 \mid Z_1,..,Z_{j-1}\right]}\right)\mathbb{P}^0\left[Z_j=1 \mid Z_1,..,Z_{j-1}\right]\right]$$
(109)

$$= \mathbb{E}^0\left[\left(\ln\left(\frac{\mathbb{P}^0\left[Z_j=0 \mid Z_1,..,Z_{j-1}\right]}{\mathbb{P}^i\left[Z_j=0 \mid Z_1,..,Z_{j-1}\right]}\right)\mathbb{P}^0\left[Z_j=0 \mid Z_1,..,Z_{j-1}\right]\right.\right.$$
(110)

$$+\ln\left(\frac{\mathbb{P}^0\left[Z_j=1 \mid Z_1,..,Z_{j-1}\right]}{\mathbb{P}^i\left[Z_j=1 \mid Z_1,..,Z_{j-1}\right]}\right)\mathbb{P}^0\left[Z_j=1 \mid Z_1,..,Z_{j-1}\right]\right)\mathbb{1}\left\{I_\epsilon^i\cap[b_3^j,b_2^j]\neq\emptyset\right\}$$
(111)

$$\left.+\int_{I_\epsilon^i}\ln\left(\frac{p^0\left[Z_j=u \mid Z_1,..,Z_{j-1}\right]}{p^i\left[Z_j=u \mid Z_1,..,Z_{j-1}\right]}\right)p^0\left[Z_j=u \mid Z_1,..,Z_{j-1}\right]du\mathbb{1}\left\{I_\epsilon^i\cap[b_3^j,b_2^j]\neq\emptyset\right\}\right]$$
(112)

Let us denote $Y_j(\mathbf{b}^j)$ is the function which has values 1 when $\beta_2^j\leq b_3^j$ and value 0 when $\beta_2^j > b_2^j$. And let us use the inequality between the KL divergence and $\chi^2$ distance.

$$KL\left(\mathbb{P}^i_{Z_j|Z_1,..,Z_{j-1}}; \mathbb{P}^0_{Z_j|Z_1,..,Z_{j-1}}\right) \leq \mathbb{E}^0\left[\left(\ln\left(\frac{\mathbb{P}^0\left[Y_j(b^j)=0\right]}{\mathbb{P}^i\left[Y_j(b^j)=0\right]}\right)\mathbb{P}^0\left[Y_j(b^j)=0\right]\right.\right.$$

$$+\ln\left(\frac{\mathbb{P}^0\left[Y_j(b^j)=1\right]}{\mathbb{P}^i\left[Y_j(b^j)=1\right]}\right)\mathbb{P}^0\left[Y_j(b^j)=1\right]\right)\mathbb{1}\left\{I_\epsilon^i\cap[b_3^j,b_2^j]\neq\emptyset\right\}$$

$$\left.+\int_{I_\epsilon^i}\frac{\left(f_2(u)-f_{2,\epsilon}^i(u)\right)^2}{f_{2,\epsilon}^i(u)}du\mathbb{1}\left\{I_\epsilon^i\cap(b_3^j,b_2^j]\neq\emptyset\right\}\right]$$

$$\leq\left(\max_{x\in I_\epsilon^i}\frac{\left(F_2^0(x)-F_2^i(x)\right)^2}{F_2^i(x)\left(1-F_2^i(x)\right)}+6\epsilon^2\right)\mathbb{P}^0\left(I_\epsilon^i\cap(b_3^j,b_2^j]\neq\emptyset\right)$$

$$\leq\epsilon^2(81+6)\mathbb{P}^0\left(I_\epsilon^i\cap(b_3^j,b_2^j]\neq\emptyset\right)$$
(113)

We can now use (100) and (113) to get :

$$\mathbb{P}^i\left[(b_3^t,b_2^t](Z_1,..,Z_{t-1})\cap\mathcal{I}_i^T\neq\emptyset\right] - \mathbb{P}^0\left[(b_3^t,b_2^t](Z_1,..,Z_{t-1})\cap\mathcal{I}_i^T\neq\emptyset\right] \leq 10\epsilon\sqrt{\mathbb{E}^0\left[N_t(i)\right]}$$
(114)

Reinjecting in (96) allows us to recover :

$$\mathbb{E}^i\left[N_T(i)\right] - \mathbb{E}^0\left[N_T(i)\right] \leq 10T\epsilon\sqrt{\mathbb{E}^0\left[N_T(i)\right]} \tag{115}$$

$\square$

## D   Instance specific proofs

Here, we present proofs and results that are specific to certain instances of the auction problem, such as $\Delta$-separated distributions and the i.i.d. unit-demand adversary setting. These specialized analyses provide further insight into the behavior of learning algorithms under specific structural assumptions.

We begin by providing the proof of Lemma 4, which we first restate for convenience.

**Lemma 4.** *Learning in a discriminatory or uniform auction when the adversary's distribution is $\Delta$ separated, with $\Delta < \frac{1}{2K}$ can be achieved with respectively regret of $\Omega(T^{2/3})$ and $\mathcal{O}(\sqrt{T})$.*

*Proof.* We first provide the proof of the lower bound in the discriminatory auction. We begin by providing the necessary tools for the reader to understand the reduction of the case of $\Delta$-separated distribution with $\Delta \leq \frac{1}{2K}$. Our first remark is the following: while the proof of Lemma 3 makes use of an adversary distribution whose support is $[0,1]$, one can easily provide another difficult instance for adversary bids on $[0,\frac{1}{2}]$. Indeed, consider, with the notations used in the proof, $\tilde{F}(b) = \max(1, F(2b))$, and a valuation $v = \frac{1}{2}$. Then the bidder faces a problem identical to the original one, scaled down on $[0,\frac{1}{2}]$, and the lower bound holds.

All that is left to show for the lower bound to extend to $\Delta$ sepearted distribution is that there exists such instances that reduces to the above *scaled down* difficult instance.

With this in mind, consider the multi-unit discriminatory auction, when the bids of the adversary are such that $\beta := (1, 1 - \frac{1}{2K}, \ldots, 1 - \frac{K-2}{2K}, b)$ where $b$ follows the distribution corresponding to the scaled down first price auction mentioned above. Note that this constitutes a $\Delta$-separated distribution for $\Delta \leq \frac{1}{2K}$. Let us consider the case when the bidders value $\mathbf{v} \in B$ is such that $v_1 \leq \frac{1}{2}$. It is straightforward when writing the utility function that for any bidder such that $v_1 \leq \frac{1}{2}$, it is strategy dominant to only emit bids smaller or equal to $\frac{1}{2}$. Therefore, from the bidder's perspective, this reduces to a first-price auction problem, with opposing distribution $\tilde{F}(b)$ and valuation $v_1$. The lower bound of $\Omega(T^{2/3})$ from Lemma 3 therefore extends to this setting for the discriminatory auction.

We now move to providing an upper bound for the uniform price auction.

We begin by showing that if the opposing bids follow a $\Delta$ separated distribution, then there exists a utility maximizing bid $\mathbf{b}^\star \in B$ such that $\mathbf{b}^\star$ values are only 0 and/or 1. Let $\mathcal{D}$ be a $\Delta$ separated distribution. For convenience, let us rewrite the utility :

$$\begin{aligned}
u(\mathbf{b}, \boldsymbol{\beta}) = &\sum_{i=1}^{K} \mathbb{1}\left\{b_i \geq \beta_{K-i+1} > b_{i+1}\right\}\left(\sum_{j=1}^{i} v_j - \beta_{K-i+1}\right) \\
&+ \sum_{i=1}^{K} \mathbb{1}\left\{\beta_{K-i} > b_{i+1} \geq \beta_{K-i+1}\right\}\left(\sum_{j=1}^{i} v_j - b_{i+1}\right)
\end{aligned} \tag{116}$$

.

Since $\mathcal{D}$ is a $\Delta$ separated distribution, given any $\mathbf{b} \in B$, there is at most one $i \in [K]$ such that $\mathbb{P}(b_i \geq \beta_{K-i+1} > b_{i+1})$ and/or $\mathbb{P}(\beta_{K-i} > b_{i+1} \geq \beta_{K-i+1})$ can be nonzero. Furthermore, notice

that given $i \in [K]$, we always have

$$\mathbb{E}\left[\mathbb{1}\{b_i \geq \beta_{K-i+1} > b_{i+1}\}\left(\sum_{j=1}^{i} v_j - \beta_{K-i+1}\right)\right.$$

$$\left. + \mathbb{1}\{\beta_{K-i} > b_{i+1} \geq \beta_{K-i+1}\}\left(\sum_{j=1}^{i} v_j - b_{i+1}\right)\right] \qquad (117)$$

$$\leq \mathbb{E}\left[\left(\sum_{j=1}^{i} v_j - \beta_{K-i+1}\right)\right]$$

.

The previous equation is obtained by noticing that $\{\beta_{K-i} > b_{i+1} \geq \beta_{K-i+1}\}$ and $\{b_i \geq \beta_{K-i+1} > b_{i+1}\}$ are disjoint and the second indicator includes the condition $b_{i+1} \geq \beta_{K-i+1}$.

Therefore, by defining $i^\star := \operatorname{argmax}_{i \in [K]} \mathbb{E}\left[\sum_{j=1}^{i} v_j - \beta_{K-i+1}\right]$. We ensure that for any $\mathbf{b} \in B$, we have $\mathbb{E}[u(\mathbf{b}, \boldsymbol{\beta})] \leq \mathbb{E}\left[\sum_{j=1}^{i^\star} v_j - \beta_{K-i^\star+1}\right]$.

Now, by definition of $\mathcal{I}_{K-i^\star+1}$, if $\mathbf{b} \in B$ is such that $\mathcal{I}_{K-i^\star+1} \subseteq (b_{i^\star+1}, b_{i^\star}]$, then $\mathbb{E}[u(\mathbf{b}, \boldsymbol{\beta})] = \mathbb{E}\left[\sum_{j=1}^{i^\star} v_j - \beta_{K-i^\star+1}\right]$, which is the maximum utility the learner can achieve. Notably, this means that one can always construct an optimal bid $\mathbf{b} \in B$ that consists only of 0s and 1s.

We can further notice from the formula of $(\tilde{F}_k)_{k \in [K]}$ in (??) that if $\mathcal{D}$ is $\Delta$ separated the empirical distribution described by $(\tilde{F}_k)_{k \in [K]}$ is also $\Delta$ separated.

Let us now suppose that, in Algorithm 3, when there is a tie between several maximizers, the algorithm chooses, if possible, bids with values 0 or 1.

With this in mind and the fact that the estimates built for the algorithm $(\tilde{F}_k)_{k \in [K]}$ necessarily correspond to a $\Delta$ separated distribution, it should be clear that, at any time $t$, the algorithm only plays bids composed of 0 and 1. Because there are only $K$ such bids, the bidder essentially plays a $K$-armed bandit problem.

Furthermore, the concentration bounds developed in the proof of Theorem 4 ensure with probability $1 - 2K\alpha$

$$\sup_{\mathbf{b} \in \prod_{i \in [K]} \mathcal{I}_k^t} \left| \mathbb{E}_{\boldsymbol{\beta} \sim \mathcal{D}}(\mathbf{b}, \boldsymbol{\beta}) - \tilde{u}^t(\mathbf{b}) \right| \leq 6K^{3/2}\epsilon \qquad (118)$$

And therefore, if we denote $\Delta_i^\star := \min_{i \in [K], i \neq i^\star} \mathbb{E}\left[\sum_{j=1}^{K} v_j - \beta_{K-i^\star+1}\right] - \mathbb{E}\left[\sum_{j=1}^{K} v_j - \beta_{K-i+1}\right]$, as soon as $6K^{3/2}\epsilon \leq \Delta_i^\star$, the bidder only plays the optimal arm.

The regret therefore scales as $\tilde{\mathcal{O}}(\sqrt{T})$ in an instance dependent fashion (depending on $\Delta_i^\star$).

$\square$

We now move to providing the postponed proofs for the unit-demand symmetric case. We begin by proving our concentration result Lemma 5.

**Lemma 5.** *Let $t, k \in \mathbb{N}$, and let $X_j^i$ for $i \in [t], j \in [k]$ be i.i.d. samples from a distribution $\mathcal{P}$ with cdf $F$. With probability $1 - \alpha$ :*

$$\hat{F}_{k \to k'}^t(x) - \alpha_{k,k'}\epsilon \leq F_{k'}(x) \leq \hat{F}_{k \to k'}^t(x) + \alpha_{k,k'}\epsilon \qquad (8)$$

*Where $\epsilon = \sqrt{\frac{\ln\left(\frac{2}{\alpha}\right)}{2t}}$ and $\alpha_{k,k'}$ is a constant which only depends on $k, k'$ and $N$.*

*Proof.* Let $k, k' \in [K]^2$, and $t \in [T]$ such that one has observed $X_{(k)}^1, \ldots, X_{(k)}^t$, the $k^{\text{th}}$ ordered statistic $t$ times. We can apply the DKW inequality from Massart, 1990 to bound the distance from its empirical CDF to its actual CDFs denoted $\hat{F}_{(k)}$ and $F_{(k)}$. This yields the following

$$\mathbb{P}\left(\sup_{x \in [0,1]} |\hat{F_{(k)}}^t(x) - F_{(k)}(x)| \leq \sqrt{\frac{\ln\left(\frac{2}{\alpha}\right)}{2t}}\right) \geq 1 - \alpha \tag{119}$$

We denote $\epsilon = \sqrt{\frac{\ln\left(\frac{2}{\alpha}\right)}{2t}}$ Using (7), one has that for $F$ the CDF corresponding to $\mathcal{P}$, $P_k(F) = F_{(k)}$, where $P_k$ is the polynomial defined by (7). This polynomial is strictly increasing on $(0, 1)$, we can therefore get the following inequalities with probability $1 - \alpha$, $\forall x \in [0, 1]$:

$$P_k^{-1}\left(F_{(k)}(x) + \epsilon\right) \leq F(x) \leq P_k^{-1}\left(F_{(k)}(x) + \epsilon\right) \tag{120}$$

Using (7) once again allows us to get with probability $1 - \alpha$, $\forall x \in [0, 1]$:

$$P_{k'}\left(P_k^{-1}\left(F_{(k)}(x) + \epsilon\right)\right) \leq F_{(k')}(x) \leq P_{k'}\left(P_k^{-1}\left(F_{(k)}(x) + \epsilon\right)\right) \tag{121}$$

We can then obtain concentration inequality in the usual formulation, with probability $1 - \alpha$:

$$\left|P_{k'}\left(P_k^{-1}\left(F_{(k)}(x)\right)\right) - F_{(k')}(x)\right| \leq \epsilon \alpha_{k,k'} \tag{122}$$

Where $\alpha_{k,k'}$ only depends on the upper bound of the derivatives of both $P_{k'}$ and $P_k^{-1}$ on $(0,1)$. $\quad\square$

With this, we now have the necessary tools to prove the Theorem 5. Let us first restate the theorem :
**Theorem 5.** *When facing i.i.d. adversaries in the uniform auction with bandit feedback, Algorithm 5 guarantees the regret is upper bounded by* $\tilde{\mathcal{O}}\left(\sqrt{T}\right)$.

*Proof.* As noted in Lemma 7, it is known in advance that bidding such that $b_1 = v_1$ is optimal. Since this is compatible with our Algorithm 5, we assume that amongst the maximizers of $\tilde{u}^t$, the algorithm always chooses one such that $b_1 = v_1$. Let $t \in [T]$, recall that $\bar{u}^t(\cdot) =: U_u((\bar{F}_k)_{k \in [K]}, \cdot)$ and $\mathbb{E}_{\boldsymbol{\beta} \sim \mathcal{D}}[u(\mathbf{b}, \boldsymbol{\beta})] := U_u((F_k)_{k \in [K]}, \cdot)$. Notice that the observations formalized in Lemma 2 ensure that for any $x \in [0, v_1]$, at time $t$, there exists $k'$ such that $t^{k'(x)} \geq t/K$. We can therefore leverage Lemma 5 to obtain the following for $k \in [K]$ with probability $1 - \alpha$:

$$\sup_{x \in [0,v_1]} \left|\bar{F}_k^t(x) - F_k(x)\right| \leq \epsilon^t \gamma \tag{123}$$

Where $\gamma = \max_{k,k' \in [K]^2} \alpha_{k,k'}$ and $\epsilon^t := \sqrt{\frac{\ln\left(\frac{2}{\alpha}\right)}{2t/K}}$.

Note that we can ensure the bound is true across all $k \in [K]$ with probability $1 - K\alpha$

Reinjecting directly and using the formula from (15), one gets the following : with probability $1 - K\alpha$ :

$$\sup_{\substack{\mathbf{b} \in B \\ b_1 = v_1}} \left|\bar{u}^t(\mathbf{b}) - \mathbb{E}_{\boldsymbol{\beta} \sim \mathcal{D}}[u(\mathbf{b}, \boldsymbol{\beta})]\right| \leq 3K^2 \epsilon^t \gamma \tag{124}$$

Where $\gamma = \max_{k,k' \in [K]^2} \alpha_{k,k'}$ and $\epsilon^t := \sqrt{\frac{\ln\left(\frac{2}{\alpha}\right)}{2t/K}}$.

Using the optimality of $\mathbf{b}^t$ with respect to $\bar{u}^t(\mathbf{b})$, we get that with probability $1 - K\alpha$:

$$\mathbb{E}_{\boldsymbol{\beta} \sim \mathcal{D}}[u(\mathbf{b}^t, \boldsymbol{\beta})] \geq \mathbb{E}_{\boldsymbol{\beta} \sim \mathcal{D}}[u(\mathbf{b}^\star, \boldsymbol{\beta})] - 6K^2 \epsilon^t \gamma \tag{125}$$

Summing over $t \in [T]$ and taking $\alpha = \frac{1}{T^3}$ yields the desired regret bound.

$\square$

We finally provide the reduction necessary to obtain the lower bound of Lemma 6, which we restate first :

**Lemma 6.** *When facing i.i.d. adversaries in the uniform auction with bandit feedback, any learning algorithm must incur at least a regret of $\Omega(\sqrt{T})$.*

*Proof.* We focus on showing that we can build a reduction to one of the *hard instances* used in the proof of the full-information feedback lower bound in Brânzei et al., 2023. We focus on the 2 item auction.

Let $p = \frac{2}{3}$, if we sample two independant binomial random variable $X_1, X_2$ with probability $p$ multiply them by $\frac{2}{3}$ and denote that $\boldsymbol{\beta} := \left(\frac{2}{3}\max(X_1, X_2), \frac{2}{3}\min(X_1, X_2)\right)$ then with probability $\frac{1}{9}$, $\boldsymbol{\beta} = (0,0)$, with probability $\frac{4}{9}$, $\boldsymbol{\beta} = (\frac{2}{3}, 0)$ and with probability $\frac{4}{9}$, $\boldsymbol{\beta} = (\frac{2}{3}, \frac{2}{3})$. First notice that having with probability $\frac{1}{9}$, $\boldsymbol{\beta} = (0,0)$ does not change the best strategy (since the utility function $u(\cdot, (0,0))$ is constant). Then notice that the probability of $\boldsymbol{\beta} = (\frac{2}{3}, 0)$ and $\boldsymbol{\beta} = (\frac{2}{3}, \frac{2}{3})$ have derivatives in $p = \frac{2}{3}$ of opposing signs which ensures we can create perturbations of size $\delta$ in opposing direction by small changes in $p$. These are sufficient to ensure one can reproduce the same proofs as in Brânzei et al., 2023 Theorem 4 and lower bound the regret in this setting by $\Omega(\sqrt{T})$. $\qquad\square$

