# OpenReview forum: "Comparing Uniform Price and Discriminatory Multi-Unit Auctions through Regret Minimization"
_NeurIPS.cc/2025/Conference — NeurIPS 2025 poster_

### Official Review · Reviewer_mhSS · 2025-06-28

**Clarity:** 2
**Significance:** 2
**Originality:** 2
**Rating:** 3
**Confidence:** 3

**Summary:**

The authors study a "learning to bid" problem in a repeated auction of several identical units of an indivisible item. They study two possible welfare-maximizing auctions that differ in their payment rules: uniform price and discriminatory price. This problem, and the derivation of regret rates for learning to bid, has been studied recently by Branzei et al. and Galgana and Golrezaei. The key conceptual contribution of the present work is to consider a stochastic model of utilities, where the bidder of interest evaluates her utilities in a Bayesian fashion over the draw of the other competing bids. Regret is also evaluated with respect to the stochasticity in the other bids. This allows the authors to transfer over many of the previous regret rates. They also show the existence of settings where learning to bid in uniform price auctions is easier than in discriminatory price auctions.

**Questions:**

Line 15: Perhaps that claim would have stronger support if the authors included a reference that is more recent than 2009.

Lines 61-71: regret rates are presented with dependence on K, but K has not once been defined before that.

Line 99: I think you should explicitly state immediately that the K items are K identical copies of the same indivisible item.

Line 104: If the \\(v_{\ell}\\) are marginal valuations, is there an assumption that \\(v_1\ge v_2\ge\cdots\ge v_K\\)? In Line 110, are the beta's also drawn from a distribution such that the beta's have decreasing marginal utilities?

Line 119: I believe that the decreasing marginal returns assumption above is required for this to be the correct allocation function that implements the efficient/welfare-maximizing allocation. The authors should make that clear. Also, it might help to (i) use the words "efficient" or "welfare-maximizing" instead of "standard" (line 118), and (ii) include a one line explanation why the allocation function has that particular form.

Lines 124-126: "(the bid that won its emitter this item)" I don't understand what this means. Based on eqn (3), the discriminatory price auction sets the price of the kth item to b_k (the bidder's marginal value for the kth unit), regardless of who wins. Is that correct? I guess the b_k's for k >= x(b, beta) don't impact the bidder's utility.

Remark 1: is this well-known? If so, include a citation/reference. Otherwise a few lines of intuition would be good. I don't think the authors should include precise claims without proof.

Section 5: I was a bit confused as to how this is "beyond worst-case" as opposed to the previous section. My understanding is that this section is the same bandit-feedback setting as Section 4, and the authors are just proposing and analyzing a better algorithm (basically doing a more refined CDF estimation). (OK, I see now that the main application is for instance-dependent bounds. It would help if you included a formal discussion of the benefits instance-dependent bounds provide. I know that it's a standard thing in this literature, but putting things into context would help.)

Did Branzei et al. and Galgana+Golrezaei study separations in learning complexity between the two auction formats?

**Ethical Concerns:**

["NO or VERY MINOR ethics concerns only"]

**Final Justification:**

I will maintain my score as borderline as I am not entirely able to see a sufficient conceptual advancement over prior works--the main contribution appears to be a modified setting that allows the two formats to be studied in a unified sense.

**Limitations:**

Yes.

**Paper Formatting Concerns:**

Formatting issues with abstract: no "Abstract" title, and margins are too small.

**Quality:**

3

**Strengths And Weaknesses:**

Strengths
---

Multi-unit auctions are prevalent in the real world, and learning to bid in these repeated auctions is an important problem that has received study in recent years. The authors contribute to that literature with a model of stochastic adversaries, showing that many of the regret bounds transfer over to their setting, and derive improved results in some cases.

The standalone improvement (Lemma 3) to the regret lower bound of Galgana+Golrezaei is nice, and the setting showing that learning to bid in uniform price auctions is strictly easier (O(T^{1/2}) vs. Omega(T^{2/3})) than in discriminatory price auctions, is fairly interesting.

Weaknesses
---

I found it difficult to assess the contribution over the prior works of Branzei et al. and Galgana and Golrezaei. The authors should improve the presentation to emphasize the main differences---both conceptual and technical. From what I could gather, the model is identical to those two papers, and the only difference is in how the other competing bidders are treated. In the present work, the competing bidders are treated as stochastic, and the main bidder's objective is to maximize her expected utility over the draw of the competing bids. The main techniques appeal to standard CDF estimation concentration bounds. Here, some differentiation over the techniques from prior work would help the reader substantially. Also, as  I mention below in the comments section, some further exposition clarifying the purpose of instance-dependent guarantees (and what exactly they mean mathematically) would be helpful.

The proof sketch that involves multiclass classification/Natarajan dimension sounds quite interesting to me and I would have liked to see a slightly more-expanded intuitive explanation in the main body. Where/why does multiclass classification come into the picture?

---

> ### Author Rebuttal · Authors · 2025-07-30
>
> Thank you for the review.
>
> ## Comparison with prior work
> We begin by clarifying how our work differs from previous ones (Potfer et al., Branzei et al. and Galgana et al.).
>
> ### Comparing auction formats
>  All these previous works focused primarily on developing learning algorithms for one auction format at a time and studied the case of adversarial opposing bids. This work studies and compares learning in both the auction formats and focuses on stochastic opposing bids.
> Galgana et al. provide some numerical experiments comparing both formats in how quickly the bids converge and regarding the equilibrium achieved, in the appendix without theoretical analysis.
>
> ### Technical differences
>  We will make sure to better highlight the technical differences with prior work. Since we focus on stochastic bids, our algorithms are fundamentally different in the techniques they leverage. While other work relied on discretizing the bidding space $B$ (defined in line 109)  and applying a randomized algorithm, our algorithm works directly on $B$ and does not need randomization. As you mentioned, we mainly rely on statistical CDF estimation techniques. Besides the fact that this approach is technically different, it allows for a simpler and more tractable analysis, which is the basis allowing us to compare both auction formats under a common framework.
>
>
>
> ## Natarajan dimensions intuition
> Regarding the Natarajan/multiclass classification, we believe a good intuition for its usefulness in our special case is the following: in multiclass classification, given a context, you want to evaluate the probabilities of the sample belonging to each available class; because the classes are mutually exclusive, the probabilities must sum to one. Furthermore, when observing the true class of one sample, the indicator function corresponding to the true class takes the value 1 and the others take the value 0.
> In our problem, a very similar structure appears when, given a bid profile $\mathbf{b}$, we make estimates of the probabilities of obtaining $1,2, ... $ or $K$ items. This is similar to the multiclass setting in the sense that both probabilities and indicator functions of the events must also sum up to 1 here.
> This is the property that we mention as "strong negative correlation" on line 174.
>
>
> ## Line-by-line comments
>
>
> - Line 15:
>  You are right, adding a more recent reference will strengthen the argument. We will add the following reference :
> Khezr, Peyman, and Anne Cumpston. "A review of multiunit auctions with homogeneous goods." Journal of Economic Surveys 36.4 (2022): 1225-1247
>
> - Line 61-71:
>  This can indeed be unclear; we will ensure to mention $K$ is the number of items available at auction before using the notation.
>
> - Line 99:
>  We will note right away that items are indivisible and identical to ensure the setting is clear.
>
> - Line 104:
>  The $(\beta_k)_{k\in [K]}$ are indeed non-increasing (they belong to $B$, which is defined on line 109 as the non-increasing sequences of $[0,1]^K$). It is indeed also common to assume the marginal valuations $v_1,...,v_K$ are non-increasing. This question was also raised by another reviewer, we will make this more explicit in the final version.
>
>
>
> - Line 119:
> We used the term "standard" instead of "efficient" or "welfare maximizing" as these two types of auctions have been studied and shown to be inefficient by De Keijzer, Bart, et al. and they are called "standard auction" in this work (cf [1] end of this response).
>  We nevertheless acknowledge that the term "standard" might not be the most transparent and will prefer a descriptive approach explaining the rationale behind the allocation, highlighting the fact that it would be optimal if bidders were to bid truthfully (bidding $b_i=v_i$).
>
>
>
>
> - Line 124-126:
> Your understanding is mostly correct. In the discriminatory auction, the price of the $k^{th}$ item \textit{allocated to the bidder} is set as $b_k$. Be aware that the bid $b_k$ is not necessarily the bidder's marginal valuation $v_k$. Since these auction formats are non-truthful, it is not optimal to bid $b_i=v_i$.
> You are correct in your understanding that the $b_k$ for $k > x(\mathbf{b},\beta)$ do not change the bidder utility.
>
>
> ## Remark 1
>  We understand the need for a more detailed argument. As it is not the main focus of the paper, and to adhere to space constraints, we did not provide too much detail.
> The main arguments to show the complexity of the maximization are the following :
> - $\hat{u}^t (\mathbf{b})$ is written as a function of $\mathbf{b}$ and $\hat{F}_k^t$ (equations (13) and (14)) and it is separable into functions of neighbouring pairs of components of $\mathbf{b}$.
> - Each of these functions is piecewise linear with pieces joining at the past observed $\beta_k^t$.
>
> Given these two points, one can prove the complexity bounds by using Theorem 1 in Branzei et al., which shows that the maximization problem is equivalent to a shortest path problem in a Directed Acyclic Graph.
>
>
>
> ## Section 5
>  This section indeed focuses on Bandit feedback. The main point of this section is the following: in general, we cannot say one auction or the other is easier to learn; yet if we restrict the possible distributions (or the valuation of the bidder), then some results showing that both auctions are not always equivalent can be obtained. This section focuses on showcasing some of these cases and providing the corresponding analysis.
>
> ## Instance Dependent bounds
> The strength of instance-dependent bounds is that they represent how the regret can be lower for certain instances. In short, they allow us to characterize what parameters make the learning problem easier or harder and how much our algorithms can leverage them. In our case, that can be having only unit-demand (or two-unit demand) or having an opposing bids distribution that is well-behaved. It is the focus of section 5 to understand what these parameters are for our auction settings and how they might differ depending on the auction format.
> We will make sure to more clearly explain why it is beyond worst case and highlight the interest of these bounds in the paper.
>
>
>
>
>
> [1] De Keijzer, Bart, et al. "Inefficiency of standard multi-unit auctions." European Symposium on Algorithms. Berlin, Heidelberg: Springer Berlin Heidelberg, 2013.
>
> [2] Potfer, M., Baudry, D., Richard, H., Perchet, V., & Wan, C. (2024). Improved learning rates in multi-unit uniform price auctions. Advances in Neural Information Processing Systems, 37, 130237-130264.
>
> [3] Brânzei, S., Derakhshan, M., Golrezaei, N., & Han, Y. (2023). Learning and collusion in multi-unit auctions. Advances in Neural Information Processing Systems, 36, 22191-22225.
>
> [4] Galgana, R., & Golrezaei, N. (2025). Learning in repeated multiunit pay-as-bid auctions. Manufacturing & Service Operations Management, 27(1), 200-229.

---

> ### Comment · Reviewer_mhSS · 2025-08-03
>
> Thank you for your response.

---

### Official Review · Reviewer_AAkx · 2025-06-29

**Clarity:** 4
**Significance:** 3
**Originality:** 4
**Rating:** 5
**Confidence:** 4

**Summary:**

The paper studies multi-unit auctions (MUAs), where several identical items are sold in each round. It focuses on two common formats: uniform-price and discriminatory-price auctions. While both allocate items to the highest bidders, they differ in how winning bidders are charged. The central question explored is how a single bidder, participating in repeated instances of such auctions, can learn to optimize their strategy when the aggregated bids of their opponents follow a known probability distribution.

The analysis is framed through the lens of regret minimization, evaluating the bidder’s ability to learn over time. Two feedback settings are considered: full information and bandit feedback. In the full information setting, both auction types pose similar learning challenges. However, under bandit feedback, the paper shows that uniform-price auctions allow for a smaller instance-dependent regret than discriminatory auctions, making them more favorable for learning in practice. While most of the novel theoretical results concern uniform-price auctions, the key contribution of the paper lies in the comparative perspective on the learning complexity in the two auction formats.

**Questions:**

Can you comment on the dependence of the regret on specific parameters of the instance for the instance-dependent bound ?

**Ethical Concerns:**

["NO or VERY MINOR ethics concerns only"]

**Final Justification:**

After having read responses to my review and others', I still have a positive opinion of the contributions, though maybe some light edits/clarifications would be beneficial to the paper.

**Limitations:**

yes

**Quality:**

3

**Strengths And Weaknesses:**

**Strengths :** The motivation is clear and straightforward. The topic itself is timely and interesting, addressing a well-established yet practically significant problem in auction theory. The paper is well-written, with a coherent structure and precise explanations that guide the reader through many different instances of the problem.. The results presented are thorough and cover a wide range of scenarios. Moreover, the contributions are not only substantial in number but also of high quality.

**Weaknesses :** While the paper is strong, there are a few weaknesses that could be improved. Some parts of the general setting are not entirely clear. For example, it is not stated whether the bids $\beta_1^t, \ldots, \beta_n^t$ are ordered in increasing value, which can affect the interpretation. The issue of non-truthfulness is also not discussed in depth. This is important because, in single-unit auctions, uniform-price formats are known to be truthful. The special case where K=1 is not clearly separated from the general case, which makes the analysis feel less complete. For the instance-dependent regret results, it would have been helpful to study how the regret depends on specific parameters of the instance, to better understand which situations are easier or harder. Finally, there are no experiments included in the paper, but this is acceptable given the theoretical nature of the contributions.

---

> ### Author Rebuttal · Authors · 2025-07-30
>
> Thank you for the review.
>
> ## Question
> We characterize how the regret scales differently according to the number of positive valuations of the items (unit demand, two-unit demand, and more ). This is shown in Table 1 under the "worst case" column.
> While we did investigate further how specific parameters of the instance change the instance-dependent bound, our analysis only yielded conditions on the expected utility function. We were not able to derive explicit conditions on parameters of the instance (distribution, valuations, etc.).
>
> ## Non-truthfullness
> The two auction formats we consider are non-truthful. It is not clear to us how we could discuss further the non-truthful aspect of these auction formats, as the paper looks into how to learn online the best bids, which wouldn't be needed if the auction were truthful.
> We nevertheless included Lemma 7 in the Appendices, which describes how the uniform price auction is truthful for the first value/bid. We will consider making this aspect clearer in the main paper.
>
> ## Single unit auction
>  Regarding the K=1 special case (for which our comparison reduces to a first-price versus second-price auction), it is indeed not addressed on its own. Nevertheless, it is a special case of the \textit{unit demand} setting for which some results are mentioned (lines 285-288). Indeed, if K=1, then the bidder only has $v_1>0$
>
> ## Opposing bids
>  You are right, $\beta^t_1, ..., \beta^t_K$ are indeed in non-increasing order. This is expressed in the paper by the fact that they belong to $B$, defined on lines 107-109 as the subset of $[0,1]^K$ composed of the non-increasing sequences. We note that this is somewhat implicit and will try to highlight it more explicitly.

---

> > ### Comment · Reviewer_AAkx · 2025-08-06
> >
> > Thank you for your answers.
> >
> > I apologize for missing the fact hat $\beta_i^t$ belong to $B$ and are thus non-increasing.
> >
> > About non-truthfulness, I merely meant that I think the paper could be more precise about non-truthfulness (not valid for uniform price for first bid e.g.).
> >
> > After having read responses to my review and others', I still have a positive opinion of the contributions, though maybe some light edits/clarifications would be beneficial to the paper.
> >
> > I am therefore enclined to maintain my grade.

---

### Official Review · Reviewer_4GPy · 2025-06-30

**Clarity:** 1
**Significance:** 3
**Originality:** 2
**Rating:** 3
**Confidence:** 3

**Summary:**

The authors study multi-unit uniform price and discriminatory price auctions from an online learning perspective. At each time step $t$, the learner participates in either a uniform (respectively, discriminatory) auction with the goal of winning up to $K$ identical items by submitting a vector of bids $(b_1^t, …, b_K^t)$. It is assumed that the learner’s utility is additive and that the utility associated with winning the $k$-th item for any $k \in [K]$ is known and fixed over time.
The learner’s bids compete with the top $K$ bids submitted by other participants. The learner wins exactly $k$ items if precisely $k$ of their bids are among the $K$ highest overall. The payment scheme is as follows: in uniform price auctions, the learner pays $k \cdot p$, where $p$ is the value of the first rejected bid; in discriminatory price auctions, the learner pays the sum of their $k$ winning bids.
The feedback model can be either full-information—where the learner observes all the top $K$ competing bids—or bandit feedback, where the learner only observes the number of items won and the corresponding price. The competing bids are modeled as independent and identically distributed (i.i.d.) random variables.
The learner’s objective is to minimize regret with respect to a benchmark bidder who knows the distribution of the competing bids and bids optimally to maximize expected reward.
As far as I can tell (despite several typos in the paper that make this somewhat unclear) these appear to be the authors’ main contributions:
1.	Full-feedback: The authors claim to improve the previously known upper bound for uniform price auctions by a $\sqrt{K}$  factor. However, I believe there may be a typo here. In the related work section, they state that Branzei et al. achieve a regret rate of $O(K \sqrt{T})$, which is exactly the rate the authors obtain in this paper.
It is unclear whether the authors provide any additional improvements for the full-feedback setting. They state: "We show that both auction formats admit tight worst-case regret rates of $\tilde{O}(K\sqrt{T}))$ under the full-information setting (improving the best known rates for uniform price auctions by a factor of $\sqrt{K}$)." This wording suggests they also contribute something for discriminatory price auctions, but I did not manage to find any explicit contribution in this sense, leaving the reader uncertain about what has actually been improved.
2.	Bandit Feedback – Upper and Lower Bounds: For the bandit feedback setting, the authors claim: "Our analysis also yields regret rates of $\tilde{O}(K^{5/3} T^{2/3})$ for uniform price auctions, and we show a matching lower bound."
However, this seems problematic. Based on the context, this result should only hold in the stochastic i.i.d. setting. Furthermore, the authors reference Potfer et al., who achieve a regret rate of $\tilde{O}(K^{4/3} T^{2/3})$, which seems actually better than the upper bound presented in this paper. If Potfer et al.'s result is correct, it would contradict the claim that the authors' lower bound is tight. In fact, I could only find a $\Omega(T^{2/3})$ lower bound in the paper, not the $\Omega(K^{5/3} T^{2/3})$ bound they seem to suggest with the phrasing above. Overall, this part of the contribution is unclear.
3.	Instance-Dependent Rates: The authors also claim to provide improved instance-dependent regret rates for the uniform price auction, specifically for one-unit and two-unit demand cases, along with matching lower bounds in the time horizon.

**Questions:**

Could you please clarify what are the contributions relative to the literature answering to the point raised in the summary?
Also, the summary table 1 should report also the dependences on K, not just on T.
Finally, a "techniques and challenges" section would be beneficial for the reader to understand better what are the non-standard technical contributions with respect to the existing literature.

Typo: line 123, diplayed formula: extra ]

I am willing to reconsider my score depending on the authors response.

**Ethical Concerns:**

["NO or VERY MINOR ethics concerns only"]

**Final Justification:**

I still believe that the paper is slightly below the acceptance threshold: in the light of the further discussion about the related work provided by the authors in the rebuttal, I believe that the actual contributions are in fact incremental (e.g., their proposed algorithm provides worst-guarantees wrt existing algorithms and only improves in distribution-dependent guarantees, and the lower bounds are not tight in $k$).

**Limitations:**

Yes

**Quality:**

2

**Strengths And Weaknesses:**

I believe that understanding the regret rates in stochastic i.i.d. uniform and discriminatory is an interesting contribution.
However, as already pointed out in the summary above, the related work and contribution section left me perplexed about how to contextualize the contributions provided by this paper.

---

> ### Author Rebuttal · Authors · 2025-07-30
>
> Thank you for the review.
>
> We take note that the exposition of the literature and of the contribution could be clearer and will improve the issues you raised.
> We first provide clarification regarding the issue raised in the summary and then answer the remaining questions.
>
>
>
> ## Typo in the literature review
> There is indeed a typo here. In the uniform price auction, under full-information feedback, Branzei et al, provided an upper bound which grows as $\tilde{\mathcal{O}} \left ( K^{3/2} \sqrt{T} \right )$, leaving a gap with the lower bound of $\Omega \left ( K \sqrt{T} \right )$. In this work, we provide an algorithm that guarantees a regret upper bounded by $\tilde{\mathcal{O}} \left ( K \sqrt{T} \right )$. In light of the previously mentioned lower bound, this algorithm's regret upper bound is optimal, fully characterizing the regret rates of this problem as $\Theta \left ( K \sqrt{T} \right )  $. We apologize for this particularly confusing typo.
>
> ##  Contribution bandit feedback
>  - Under bandit feedback, for the uniform price auction, the main contribution of our work is the lower bound of order $\Omega \left (T^{2/3} \right )$ provided by Theorem 3. In the literature, the best lower bound available for the uniform price auction was the one in the full-information setting of $\Omega \left (K \sqrt{T} \right )$ by Branzei et al. This lower bound matches the rates in $T$ of the upper bounds provided by Potfer et al., therefore fully characterizing how regret scales in $T$ in this setting.
> We will ensure to better highlight that this lower bound is tight in $T$, not in $K$.
> - As you mentioned, the worst-case bound we obtained for Algorithm 3 is worse by a factor $K^{1/3}$ in comparison to Potfer et al.. The main novelty of this algorithm is that it is able to adapt, guaranteeing $\tilde{\mathcal{O}} \left ( K \sqrt{T} \right )$ in the bandit setting when facing "easy" instances. The worst-case guarantees of $\tilde{\mathcal{O}} \left ( K^{5/3} T^{2/3} \right )$ ensure that, while able to leverage favorable instances, this algorithm still guarantees reasonable regret in other instances.
>
> ##  Techniques and challenges
> We understand the need to provide more information on how our work differs from previous ones (Potfer et al., Branzei et al., Galgana et al.) in the techniques we used for algorithm design and analysis. We will include a "techniques and challenges" subsection.
> We will emphasize the fact that since we focus on stochastic opposing bids, our algorithms are fundamentally different from the ones used in previous work. Namely, while the previous approaches relied on discretizing the bid space $B$ (defined on line 109) and using randomized bandit algorithms, we neither need to discretize nor use randomized algorithms. Our algorithms mainly leverage statistical CDF estimation techniques. A key advantage of this approach is that it allows for a more straightforward analysis.
>
>
>
>
> ##  Table 1
>  We did not specify the dependence in $K$ in Table 1 to improve readability. We were able to obtain upper and lower bounds on the regret that match in their dependence on the time horizon $T$. Yet, the dependence on the number of items, $K$, is not tight. To represent these results accurately, we would need to present upper and lower bounds separately, using $\tilde{\mathcal{O}}$ and $\Omega$ instead of $\Theta$.
> We emphasized the differences in scaling with respect to $T$, as we believe this is the most interesting aspect of the contribution.
> To ensure that the notation is clear and represents the bounds fairly, we propose to mention that $\tilde{\Theta}$ hides factors that depend on $K$.
> Would these improvements be satisfying?
>
>
> [1] Potfer, M., Baudry, D., Richard, H., Perchet, V., & Wan, C. (2024). Improved learning rates in multi-unit uniform price auctions. Advances in Neural Information Processing Systems, 37, 130237-130264.
>
> [2] Brânzei, S., Derakhshan, M., Golrezaei, N., & Han, Y. (2023). Learning and collusion in multi-unit auctions. Advances in Neural Information Processing Systems, 36, 22191-22225.
>
> [3] Galgana, R., & Golrezaei, N. (2025). Learning in repeated multiunit pay-as-bid auctions. Manufacturing & Service Operations Management, 27(1), 200-229.

---

> > ### Comment · Reviewer_4GPy · 2025-08-08
> >
> > Thank you for the clarifications. Your explanations regarding the full-information setting and the bandit feedback bounds are helpful. I will take your rebuttal into account and discuss it with the other reviewers.
> > Could you also clarify explicitly what your novel contributions are in the case of discriminatory auctions?

---

> ### Author Response · Authors · 2025-08-08
>
> Thank you for your feedback. We are happy to provide further clarification.
> Our contribution for the discriminatory auction are the following.
>
> ## An Algorithm for the stochastic opposing bids
> We propose the first algorithm for the stochastic opposing bid setting under the full-information feedback, which relies on CDF concentration bands. The main upsides of this algorithm compared to the one available in Galgana and Golrezaei 2025  are that it is deterministic, doesn't use a discretized version of the action space $B$, and its analysis is short and straightforward.
>
>
> ## A More general lower bound for bandit feedback
> We also provide an improved version of the existing lower bounds of $\Omega ( T^{2/3} )$ for the bandit feedback. We improve on the already existing bound in the following sense : the previously known lower bound scaled similarly with $T^{2/3}$ but was only valid for bids which can only take a discete set of values. We provide a version which allows for the bids to take any values in $B$. In essence, previous bounds only applied to discrete a action space while the bound we provide is valid for a continuous one (which is the setting in this paper).
>
> ## Lower bounds validity for certain family of instances
> We finally show, in section 5, that under bandit feedback, the lower bound still stands in several special cases in which the uniform price auction can be learned with a regret guarantee $\tilde{\mathcal{O}} \left ( \sqrt{T} \right )$.
>
> Galgana, R., & Golrezaei, N. (2025). Learning in repeated multiunit pay-as-bid auctions. Manufacturing & Service Operations Management, 27(1), 200-229.

---

### Official Review · Reviewer_Wq2f · 2025-07-04

**Clarity:** 4
**Significance:** 2
**Originality:** 2
**Rating:** 4
**Confidence:** 4

**Summary:**

The paper compares two common multi-unit auction formats—uniform-price and discriminatory auctions—from the perspective of a bidder using online learning to optimize strategies against stochastic adversaries. It analyzes regret bounds under full-information and bandit feedback settings, highlighting both worst-case and instance-dependent learning difficulties.

**Questions:**

NA

**Ethical Concerns:**

["NO or VERY MINOR ethics concerns only"]

**Final Justification:**

I shall keep my score.

**Paper Formatting Concerns:**

No formatting issue.

**Quality:**

3

**Strengths And Weaknesses:**

Strengths:
1. This paper provides concrete and thorough theoretical proof, showing the regret bounds for uniform and discriminatory multi-unit auctions with various formats.
2. The paper is well-written and properly organized.
3. It is interesting to consider the so-called "instance-independent" regret, providing insights how the algorithms can be improved beyond worse-case.

Weaknesses:
1. The comparison results in some sense are not surprising.
2. The assumption that the bids of the adversary are drown from fixed distribution is kind of strong.

---

> ### Author Rebuttal · Authors · 2025-07-30
>
> Thank you for the review.
>
> In the absence of questions, we will address the two weaknesses mentioned in the review.
>
> 1. We believe our results are non-trivial and several aspects are surprising. For instance, the degeneracy of achievable regret rates in the uniform price auction, depending on whether the learner has unit, two-unit, or more than two-unit values (cf Table 1) is, in our opinion, rather surprising. We also consider it surprising that, under bandit feedback, when opposing bids are order statistics, uniform price and discriminatory auction exhibit regret rates which scale differently with $T$ (respectively $\Theta \left ( \sqrt{T} \right)$ and  $\Theta \left ( T^{2/3} \right)$). Nevertheless, we understand your comment since some results intuitively make sense, such as the fact that regret in both auction formats scales identically under full-information (as $\theta \left ( K \sqrt{T} \right )$).
>
> 2. It is true that for practical applications, such an assumption might be considered strong. On the other hand, considering that opposing bids are fully adversarial is also not realistic. We believe a realistic middle ground is to consider a contextual version of the problem: opposing bids would be i.i.d. conditionally on a context, which the bidder would observe before participating in each auction. In this work, to keep the modeling aspect straightforward, we do not consider that contextual aspect.
>
> We also want to point out the fact that the lower bounds derived in the paper are also valid for adversarial opposing bids. In that sense, only the upper bounds results in the paper are "limited" in their scope to stochastic opposing bids.
>
> Furthermore, technically, this assumption has two main upsides. First, it makes the analysis of the algorithms more tractable as we can use a deterministic algorithm.
> Second, it enables us to derive instance-dependent bounds; these types of bounds, which characterize regret beyond worst-case, are not classically obtained for adversarial settings in the bandit literature [1].
>
>
> [1] Lattimore, T.,  Szepesvári, C. (2020). Bandit algorithms. Cambridge University Press.

---

> > ### Comment · Reviewer_Wq2f · 2025-08-01
> >
> > Thanks for your response.

---

### Decision · Program_Chairs · 2025-09-17

**Decision:**

Accept (poster)

**Comment:**

The paper provides a theoretical comparison of uniform-price and discriminatory multi-unit auctions, a timely and well-motivated topic. Positive aspects included the thoroughness of the analysis and the novelty of considering instance-dependent regret. There was one major issue, raised by multiple reviewers, about the lack of clarity regarding the paper's novel contributions compared to a significant body of existing work. The authors' rebuttal did address this to a reasonable extent, and I tend to discount the concerns of the some of the reviewers who did not engage further on this topic. I'm recommending the paper as an accept, but I strongly encourage the authors to carefully incorporate the feedback about clarifying the relationship to Branzei et al, Galgana et al, etc.